# Environmental Xenobiotics and Epigenetic Modifications: Implications for Human Health and Disease

**DOI:** 10.3390/jox15040118

**Published:** 2025-07-13

**Authors:** Ana Filipa Sobral, Andrea Cunha, Inês Costa, Mariana Silva-Carvalho, Renata Silva, Daniel José Barbosa

**Affiliations:** 1Associate Laboratory i4HB—Institute for Health and Bioeconomy, University Institute of Health Sciences-CESPU, 4585-116 Gandra, Portugal; a29632@alunos.cespu.pt; 2UCIBIO—Applied Molecular Biosciences Unit, Toxicologic Pathology Research Laboratory, University Institute of Health Sciences (1H-TOXRUN, IUCS-CESPU), 4585-116 Gandra, Portugal; 3UNIPRO—Oral Pathology and Rehabilitation Research Unit, University Institute of Health Sciences (IUCS-CESPU), 4585-116 Gandra, Portugal; andrea.cunha@iucs.cespu.pt; 4Associate Laboratory i4HB—Institute for Health and Bioeconomy, Faculty of Pharmacy, University of Porto, 4050-313 Porto, Portugal; inessilvacosta@hotmail.com (I.C.); rsilva@ff.up.pt (R.S.); 5UCIBIO—Applied Molecular Biosciences Unit, Laboratory of Toxicology, Department of Biological Sciences, Faculty of Pharmacy, University of Porto, 4050-313 Porto, Portugal; 6CIIMAR—Interdisciplinary Centre of Marine and Environmental Research, Terminal de Cruzeiros do Porto de Leixões, 4450-208 Matosinhos, Portugal; 7UCIBIO—Applied Molecular Biosciences Unit, Translational Toxicology Research Laboratory, University Institute of Health Sciences (1H-TOXRUN, IUCS-CESPU), 4585-116 Gandra, Portugal

**Keywords:** environmental xenobiotics, epigenetics, DNA methylation, histone post-translational modifications, miRNAs, lncRNAs, toxicology, human health

## Abstract

Environmental xenobiotics, including heavy metals, endocrine-disrupting chemicals (EDCs), pesticides, air pollutants, nano- and microplastics, mycotoxins, and phycotoxins, are widespread compounds that pose significant risks to human health. These substances, originating from industrial and agricultural activities, vehicle emissions, and household products, disrupt cellular homeostasis and contribute to a range of diseases, including cancer and neurodegenerative diseases, among others. Emerging evidence indicates that epigenetic alterations, such as abnormal deoxyribonucleic acid (DNA) methylation, aberrant histone modifications, and altered expression of non-coding ribonucleic acids (ncRNAs), may play a central role in mediating the toxic effects of environmental xenobiotics. Furthermore, exposure to these compounds during critical periods, such as embryogenesis and early postnatal stages, can induce long-lasting epigenetic alterations that increase susceptibility to diseases later in life. Moreover, modifications to the gamete epigenome can potentially lead to effects that persist across generations (transgenerational effects). Although these modifications represent significant health risks, many epigenetic alterations may be reversible through the removal of the xenobiotic trigger, offering potential for therapeutic intervention. This review explores the relationship between environmental xenobiotics and alterations in epigenetic signatures, focusing on how these changes impact human health, including their potential for transgenerational inheritance and their potential reversibility.

## 1. Introduction

The environment plays a critical role in human health, influencing biological processes at both genetic and epigenetic levels [1]. Several environmental xenobiotics have been explored due to their potential to disrupt cellular homeostasis and contribute to disease development. These compounds include a wide range of substances such as heavy metals [e.g., lead (Pb), mercury (Hg), arsenic (As), and cadmium Cd)], endocrine-disrupting chemicals [EDCs; e.g., bisphenol A (BPA), phthalates, and dioxins], pesticides (e.g., organophosphates, glyphosate, and paraquat), various air pollutants like particulate matter [PM; e.g., diesel exhaust particles (DEPs)] and polycyclic aromatic hydrocarbons (PAHs), nano- and microplastics, and natural contaminants like mycotoxins and phycotoxins [2,3,4,5,6]. These compounds have their origin in industrial activities, agricultural practices, vehicle emissions, household products, or natural sources. As a consequence of the ubiquitous presence of these compounds in the environment, chronic exposure to even low concentrations may result in toxic, mutagenic, or teratogenic effects, and has been linked to several health conditions, including cancer [7], neurodevelopmental disorders [8], metabolic diseases [9], and dysfunction of the reproductive system [10].

While the precise mechanisms underlying the toxic effects of chronic low-dose exposure to environmental xenobiotics remain largely uncharacterised, growing evidence suggests that epigenetic alterations may play a key role in mediating these health outcomes. Epigenetics is the study of heritable but reversible modifications in gene activity that do not involve changes in the deoxyribonucleic acid (DNA) sequence, but are essential for proper cellular function [11]. The main epigenetic mechanisms are DNA methylation, histone modifications, and regulation by non-coding ribonucleic acids (ncRNAs) [12]. These epigenetic mechanisms provide a dynamic regulation that allows organisms to adapt to environmental changes. DNA methylation is the most studied epigenetic modification, and abnormal DNA methylation patterns have been implicated in different human diseases [13,14,15,16]. Moreover, histone modifications influence chromatin structure and gene transcription [17], and are implicated in cancer [18,19] and neurodegenerative/neuropsychiatric disorders [20,21,22], among others. Additionally, ncRNAs, including micro ribonucleic acids (miRNAs) and long non-coding ribonucleic acids (lncRNAs), regulate post-transcriptional gene expression and have been implicated in various diseases, including cancer [23] and neurodegeneration [24].

Recent research has demonstrated that environmental xenobiotics can alter epigenetic signatures, leading to long-term effects on gene expression and disease susceptibility. Heavy metals such as Pb [25], Cd [26], and As [27] can disrupt DNA methylation patterns, leading to changes in gene expression. EDCs, including BPA and phthalates, interfere with hormonal signalling pathways by modifying histone acetylation and DNA methylation, thereby increasing the risk of metabolic and reproductive disorders [28,29]. Pesticides and herbicides, such as organophosphates [30,31] and glyphosate [32,33], have been implicated in neurodevelopmental abnormalities and epigenetic dysregulation, particularly in early-life exposures. Additionally, air pollutants, including PM with a diameter of 2.5 µm or less (PM2.5) and PAHs, have been shown to trigger epigenetic modifications that contribute to cardiovascular and respiratory diseases [34,35]. Altogether, these findings highlight the importance of understanding how environmental xenobiotics influence disease outcomes through the modulation of epigenetic signatures.

This review explores the connection between environmental xenobiotics and epigenetics. Additionally, it also examines the consequences of these epigenetic alterations on human health, highlighting their potential transgenerational effects and reversibility.

## 2. Classes of Environmental Xenobiotics and Human Exposures

Environmental xenobiotics are synthetic or natural chemical compounds that are foreign to a biological system, often introduced into the environment through human activity, and may affect ecosystems and human health. These compounds have different sources, such as industrial processes, agriculture, pharmaceuticals, and consumer products. Therefore, the widespread use of synthetic chemical products and increased industrialization have promoted significant environmental contamination, raising concerns about the toxic long-term effects on human health.

Environmental xenobiotics can be divided into different groups, including heavy metals, EDCs, pesticides, air pollutants, nano- and microplastics (NPs and MPs, respectively), and toxins (e.g., mycotoxins, and phycotoxins), with each one exhibiting different physicochemical properties and, consequently, different outcomes to human health and environment.

### 2.1. Heavy Metals

Heavy metals are naturally occurring elements in the Earth’s crust that can pose serious risks to human health [36], with industrialization processes significantly increasing human exposure. Consequently, heavy metal contamination has become one of the most important environmental problems [37]. Examples of toxic heavy metals include Pb, Hg, As, and Cd (Figure 1A) [38].

Pb is associated with approximately 540,000 deaths annually, with the highest burden observed in developing countries. The U.S. Environmental Protection Agency (EPA) has set the maximum allowable contaminant level for Pb in drinking water at 15 µg/L [39].

Hg is a naturally occurring heavy metal, with a global production estimated at approximately 2300 tonnes per year [40]. One of its most toxic forms, methylmercury, accumulates in aquatic organisms and biomagnifies up the food chain, resulting in high concentrations in top predators such as seafood species. Individuals who consume large amounts of seafood can accumulate methylmercury in their bodies and experience its harmful health effects [41].

More generally, Hg exposure remains a public health concern. According to the World Health Organization (WHO), an estimated 310,000 to 630,000 children in the United States of America (USA) were born with blood Hg levels exceeding 5.8 µg/L, placing them at risk of adverse developmental outcomes [42]. Similarly, in Europe, it is estimated that around 200,000 children exceed the WHO-recommended limit of 2.5 µg/g for Hg concentration in hair [43].

Compounds containing As have been classified as carcinogenic to humans. Organic As has been placed in Group 2B (possibly carcinogenic to humans), while inorganic As is classified in Group 1 (carcinogenic to humans) by the International Agency for Research on Cancer (IARC) [44]. The WHO has set the maximum allowable As concentration in drinking water at less than 10 µg/L. However, more than 40 countries have reported levels exceeding this threshold, putting approximately 200 million people at risk of health issues [45]. Bangladesh and India (West Bengal) are the most affected regions, with groundwater As concentrations ranging from 0.5 to 4600 µg/L [46].

Cd is a highly reactive heavy metal with no known biological function and is classified as a human carcinogen (Group 1) by the IARC [47]. The EPA and WHO have established that Cd concentrations in drinking water should be below 3–5 µg/L [48]. However, data from Pakistan report Cd concentrations in drinking water ranging from 0.001 to 0.21 mg/L, which are significantly higher than the limits set by the WHO [49]. In Europe, Cd concentrations in drinking water were found to be 5 µg/L in Sweden, while in the Netherlands, values ranged between 0.1 and 0.2 µg/L [50]. Although Cd exposure from drinking water is relatively low, the average total daily intake of Cd from all sources (including food, water, and airborne particles) is estimated to be 30–50 µg, which is associated with a range of health issues [47].

### 2.2. Endocrine-Disrupting Chemicals

EDCs are a class of environmental xenobiotics that interfere with hormone homeostasis. They are characterised by their ability to disrupt the endocrine system by competition with endogenous steroid hormones for receptor-binding or hormone-transport proteins, or by altering the synthesis and metabolism of endogenous hormones [51,52]. Several compounds belong to this group, including BPA, phthalates, dioxins, and some persistent organic pollutants (POPs; Figure 1B).

BPA is one of the most widely produced and used chemicals worldwide, with an estimated annual production of over 6 billion pounds [53]. This compound is commonly used in the production of polymers, polyvinyl chloride (PVC) plastics, and flame retardants [54]. Food is the primary source of BPA for the general population, with estimated daily consumption levels ranging from 0.48 to 1.6 µg/kg of body weight [55]. A study conducted in Europe detected BPA in urine with a geometric mean concentration of 2040 ng/L in children and 1880 ng/L in mothers. These values were correlated with dietary intake of canned food [56]. In the Lebanese population, a mean urinary BPA level of 3670 ng/L was observed, with higher levels in older women and associated with metabolic syndrome, type 2 diabetes, hypertension, and obesity [57]. In the case of occupational exposure, a study conducted by He and colleagues [58] observed that workers involved in BPA synthesis were more exposed to the compound, with an average personal airborne concentration of 450 µg/m^3^. Additionally, nearly 90% of workers had higher levels of BPA in their urine after completing their work [58].

Phthalates are hydrophobic compounds widely used in plastic manufacturing as plasticizers to enhance the elasticity and flexibility of plastics [59]. Since they are not covalently bound to the polymer or other plastic materials, they can easily migrate from the plastic matrix into the environment [60]. Several studies monitoring human exposure have concluded that di-(2-ethylhexyl) phthalate (DEHP) metabolites were found in 75–90% of urine samples from randomly selected individuals in the general population [61,62].

POPs are a group of chemical substances defined by the Stockholm Convention (2001) as exhibiting four key characteristics: persistence, bioaccumulation, toxicity, and long-range environmental transport [63]. Many POPs also act as EDCs, interfering with hormonal systems in both wildlife and humans. Importantly, some POPs are pesticides (see Section 2.3. Pesticides), highlighting the considerable chemical and functional overlap among these environmental xenobiotics.

The initial list of twelve POPs, known as the “dirty dozen”, included several organochlorine pesticides [e.g., aldrin, chlordane, dichlorodiphenyltrichloroethane (DDT), dieldrin, endrin, heptachlor, hexachlorobenzene (HCB), mirex, and toxaphene], as well as industrial chemicals and by-products such as polychlorinated biphenyls (PCBs), polychlorinated dibenzo-p-dioxins (PCDDs), and polychlorinated dibenzofurans (PCDFs). In 2009, nine additional substances/categories were added, including alpha- and beta-hexachlorocyclohexane (α-HCH, β-HCH), lindane, chlordecone, hexabromobiphenyl, certain polybrominated diphenyl ethers (PBDEs; e.g., hexabromodiphenyl ether, heptabromodiphenyl ether), pentachlorobenzene, and perfluorooctane sulfonic acid (PFOS), its salts, and its precursor perfluorooctane sulfonyl fluoride (PFOSF) [63].

Due to their lipophilic nature, POPs accumulate in soil, water, and food, and biomagnify through the food chain [64]. This makes them a global public health concern, particularly in low- and middle-income countries with weaker regulatory frameworks (e.g., regions of Africa, Asia, and Latin America). Inadequate waste management practices, such as the open burning of garbage or e-waste, and the insufficient enforcement of environmental policies, further exacerbate human and ecological exposure [65].

Within the broad category of POPs, dioxins (including PCDDs, PCDFs, and certain PCBs) are of particular concern due to their extreme toxicity and endocrine-disrupting potential [66]. These compounds are unintended by-products of various industrial processes, and the most toxic dioxin, 2,3,7,8-tetrachlorodibenzo-*p*-dioxin (TCDD), has been extensively studied [67]. Dioxins are highly persistent, bind to organic matter in aquatic environments, and accumulate in organisms, ultimately entering the human food chain [66]. In fact, a study conducted in the USA found that food sources of dioxins were significantly higher, by 93%, than other sources [68]. While chronic low-level exposure is more common, notable high-exposure incidents have occurred, such as the poisoning of Ukrainian presidential candidate Viktor Yushchenko with TCDD [69], and an earlier case involving five workers in Vienna, Austria, in 1997, which represents the highest documented human TCDD exposure [70].

### 2.3. Pesticides

Pesticides are substances used to control or eliminate various types of unwanted organisms, including fungi, rodents, bacteria, insects, and weeds. They are extensively used in agriculture, domestic settings, and public health management. Herbicides, which are primarily used to kill or control unwanted plants, are the most widely used type of pesticide. This group includes compounds such as paraquat, glyphosate, 2,4-cichlorophenoxyacetic acid (2,4-D) (Figure 1C), atrazine, dicamba, trifluralin, metolachlor, and picloram, among others [71].

Glyphosate is an organophosphorus compound extensively used as an herbicide [72]. Although the concentrations of glyphosate and glyphosate-based herbicides (GBHs) found in the environment are relatively low, their widespread use can lead to accumulation and potential risks to animals and humans [73]. In fact, high concentrations of glyphosate have been detected in the urine and organs of farm animals and farmers [74,75,76]. Additionally, a study conducted in Europe found residues of this compound in the urine of 44% of the population, with a maximum concentration of 5 μg/L. In the USA, the results were more concerning, with 60–80% of individuals showing glyphosate residues in their urine, with a maximum concentration of 233 μg/L [77].

Paraquat is one of the most widely used herbicides due to its high efficacy in controlling weed and grass growth [78]. Most cases of paraquat exposure occur through poisoning (typically by accidental ingestion), particularly in developing countries [79]. Paraquat causes multiorgan toxicity in humans. However, it is particularly toxic to the lungs due to its accumulation through putrescine transporters, causing diffuse alveolar collapse, the infiltration of inflammatory cells into the interstitial and alveolar spaces, the proliferation of bronchial epithelial cells, excessive collagen deposition, and vascular congestion [80].

Additionally, insecticides represent another important group of pesticides, which include organochloride and organophosphorus compounds, among others. Organochlorine pesticides include compounds such as DDT, aldrin, dieldrin, endrin, chlordane, heptachlor, mirex, and toxaphene, many of which are classified as POPs (see the previous section) due to their environmental persistence and bioaccumulate effects. Despite their known risks, DDT is still used in certain regions of Africa, Asia, and Latin America for malaria vector control [65].

Organophosphates are a subclass of organophosphorus compounds extensively used as insecticides (e.g., parathion, malathion, and chlorpyrifos). They also have several other applications, including as flame retardants, in building materials, textiles, and as plasticizers [81]. Several studies have found high concentrations of parathion and malathion in water. For example, in India, concentrations of 2.618 μg/L of malathion were found in the surface water from the Ganga River in Kanpur [82]. Likewise, in Spain, extensive sampling in the Llobregat River basin revealed the presence of malathion in 54% of the collected samples, with concentrations reaching up to 320 ng/L. Moreover, out of the 43 pesticides analysed, 24 were found at concentrations exceeding the detection limits of the methods [83]. The toxic potential of these compounds is associated with neurotoxicity (particularly through acetylcholinesterase inhibition), carcinogenicity, kidney, liver and reproductive toxicity, and endocrine disruption [84].

### 2.4. Air Pollutants

Air pollutants include PM, DEPs, black carbon, and PAHs [e.g., naphthalene, anthracene, and benzo(*a*)pyrene] (Figure 2A). Other air pollutants include ozone (O_3_), nitrogen dioxide (NO_2_), sulphur dioxide (SO_2_), and carbon monoxide (CO).

PM is a heterogeneous group composed of compounds with varying size, chemical composition, concentration, surface area, and sources [85]. DEPs are a component of air PM and consist of a complex mixture of thousands of compounds, including fine particles, toxic organic materials, and vapours. They are formed by the incomplete combustion of diesel fuel from motor vehicle engines and various industries [86]. DEPs have been classified as a human carcinogen (Group 1: carcinogenic to humans) by the IARC, based on in vivo and human studies [87]. The ability of DEPs to promote adverse health effects depends on particle size and number. It is estimated that 90% of the total particles are ultrafine (a diameter smaller than 100 nm), which are the most hazardous, as they can reach target organs, including the brain [88].

To reduce the toxic effects of DEPs on the environment, renewable alternatives such as biodiesel have been proposed [89]. The chemical composition of these fuels, particularly the presence of oxygen (O_2_), helps reduce CO and total PM emissions, but increases NO_2_ emissions, which can also have toxic effects and act as ozone precursors [90], along with aldehydes and other organic compounds [91]. Additionally, although the total mass of PM is reduced, the soluble organic fraction of the emitted particles is greater than that of DEPs, which may influence biological effects and toxicity. Human studies have shown no significant difference in lung and cardiovascular function between exposure to DEPs and biodiesel exhaust particles [92]. Although renewable sources like biodiesel appear promising for environmental protection, studies on their toxicity remain limited. Current findings suggest they may cause effects similar to, or potentially worse than, those of diesel [93].

Black carbon, a component of fine PM, is generated during the partial or inefficient combustion of fuels containing carbon. This compound possesses characteristics that enhance its toxicity to humans and the environment, including high thermal stability, strong light absorption, insolubility in solvents, and weak wavelength dependence of light absorption [94]. Major sources of black carbon include transportation, open biomass burning, residential heating and cooking, and power production [95]. It contributes to various adverse health effects and atmospheric warming by absorbing solar radiation [96]. Several studies have correlated cumulative exposure to black carbon with increased mortality, particularly through respiratory system damage [97].

PAHs are a class of chemical compounds (e.g., benzo(*a*)pyrene, naphthalene, and anthracene, among others) that are classified as carcinogenic for humans [98]. They are included in the European Union (EU) and EPA priority pollutant lists due to their carcinogenic and mutagenic properties. This group includes over 100 compounds, but 16 are considered priority contaminants due to their toxicity [98,99]. PAHs typically occur as mixtures rather than individual compounds. It is estimated that the primary sources of PAH exposure are the consumption of contaminated food and water, as well as smoking [100]. For example, individuals who smoke are exposed to 15 (±9) ng of benzo(*a*)pyrene, 119 (±66) ng of phenanthrene, and 37 (±19) ng of pyrene per cigarette [101]. While the WHO has not set limits for all PAHs, it has defined a maximum limit for benzo(*a*)pyrene in drinking water at 0.7 μg/L [102]. Studies conducted in European and Canadian cities found concentrations ranging from 106.5 to 150.3 ng/L for 16 PAHs [103,104]. In the USA, carcinogenic PAH levels in drinking water ranged from 0.1 to 61.6 ng/L [105].

### 2.5. Nano- and Microplastics

NPs and MPs are plastic particles smaller than 1 μm and 5 mm, respectively [106,107]. They originate either from the fragmentation of larger plastic debris (secondary particles) or are manufactured intentionally for use in industrial applications, cosmetics, and textiles (primary particles; Figure 2B) [108]. Due to their widespread use and environmental persistence, NPs and MPs have become ubiquitous pollutants, being detected in drinking water [109], food products [110], air [111], and even human tissues, including lungs [112], and blood [113]. As such, they are classified as environmental xenobiotics.

On the other hand, NPs and MPs can adsorb and transport other environmental contaminants, such as heavy metals [114], pesticides [115], and POPs [116], enhancing their mobility and bioavailability. Studies have demonstrated that NPs can disrupt cellular homeostasis, leading to cytotoxic and genotoxic effects [117]. While their health impacts remain under investigation, growing evidence indicates that long-term exposure to NPs/MPs can induce epigenetic alterations [118].

### 2.6. Mycotoxins and Phycotoxins

Mycotoxins are secondary metabolites produced by fungi, particularly species of *Aspergillus*, *Penicillium*, and *Fusarium*, which contaminate food crops such as maize, cereals, nuts, and spices. Common mycotoxins include aflatoxins, ochratoxin A, fumonisins, deoxynivalenol, and zearalenone (Figure 2C) [119,120].

Aflatoxins are primarily produced by the fungal species *Aspergillus flavus* and *Aspergillus parasiticus*. However, other species, such as *Aspergillus ochraceus*, *Aspergillus pseudotamarii*, *Aspergillus parvisclerotigenus*, and *Aspergillus bombycis*, are also known to produce aflatoxins [121]. More than 20 aflatoxins have been discovered. Among them, aflatoxin B1 is classified as a Group 1 human carcinogen by the IARC due to its toxicity, mutagenicity, immunotoxicity, teratogenicity, and carcinogenicity [122]. Accordingly, the European Commission has established maximum permissible limits for aflatoxins in cereals and cereal-derived products of 2 μg/kg for aflatoxin B1, and 4 μg/kg for the total sum of aflatoxins B1, B2, G1, and G2 [123], below the limit established by the USA Food and Drug Administration (≤20 μg/kg) [124].

Ochratoxin A is naturally produced by species of *Penicillium* and *Aspergillus* fungi and is known for its toxicity, including teratogenic effects [125]. It is also classified by the IARC as a Group 2B carcinogen (sufficient evidence of carcinogenicity in animals but limited or inadequate evidence in humans) [126]. Furthermore, its degradation leads to the formation of several carcinogenic by-products, such as 14-(*R*)-ochratoxin A, ochratoxin-α, and ochratoxin amide, all of which contribute to ochratoxin A toxicity [127].

Fumonisins comprise dozens of compounds, mainly produced by the fungal species *Fusarium verticillioides* and *Fusarium proliferatum*. Among them, fumonisin B1 is classified as a Group 2B human carcinogen by the IARC [128].

Zearalenone and deoxynivalenol are produced by various species of *Fusarium* [5,129]. Zearalenone has been reported to have hepatotoxic, immunotoxin, and carcinogenic effects in animal experiments [130], while the acute effects of deoxynivalenol mainly include nausea, vomiting, and gastroenteritis [131]. However, long-term exposure to contaminated food with deoxynivalenol may cause developmental delays and growth retardation in children [131].

Fusaric acid is a secondary metabolite produced by various *Fusarium* species, including *Fusarium verticillioides* and *Fusarium oxysporum* [132]. Although less toxic than major mycotoxins, it exhibits neurotoxic, immunosuppressive, and phytotoxic properties [132,133,134].

On the other hand, phycotoxins are toxic compounds produced by harmful algal blooms, including dinoflagellates, diatoms, and cyanobacteria. They can be classified in distinct groups: microcystins, azaspiracids, brevetoxins, cyclic imines, okadaic acid and dinophysistoxins, pectenotoxins, yessotoxins, saxitoxins, and domoic acid. Although microcystins are more specifically categorized as cyanotoxins due to their cyanobacterial origin, they are also considered phycotoxins under the broader definition that includes all toxins produced by aquatic photosynthetic organisms. Some phycotoxins can contaminate water bodies and seafood, posing health risks to humans upon ingestion (Figure 2D) [6,135].

Microcystin-LR is one of the most toxic and widely studied variants of microcystins, a family of cyclic heptapeptides produced by freshwater cyanobacteria, including *Microcystis*, *Anabaena*, *Nostoc*, and *Planktothrix* species, among others [136]. This toxin poses a significant risk to both human and environmental health due to its high stability, bioaccumulation, and potent hepatotoxicity [137]. As such, the WHO has set a provisional guideline value of 1 µg/L for Microcystin-LR in drinking water. Moreover, the IARC has classified Microcystin-LR as possibly carcinogenic to humans (Group 2B) [138].

Okadaic acid is another well-known phycotoxin, produced by marine dinoflagellates such as *Dinophysis* spp. and *Prorocentrum* spp. [139]. It acts as a specific inhibitor of serine/threonine protein phosphatases and is the primary agent responsible for diarrhetic shellfish poisoning [139]. The EU recommends a limit of 160 µg of okadaic acid/kg shellfish meat [140]. In addition to its acute gastrointestinal effects, okadaic acid is also recognised as a tumour promoter in human cells and in rodents, although its direct link to human carcinogenesis needs a more exhaustive investigation [139,141]. Nevertheless, despite its tumour-promoting activity, okadaic acid has not been officially classified as a carcinogen by the IARC [139].

Growing evidence indicates that mycotoxins and phycotoxins can induce epigenetic alterations [142,143,144,145,146,147,148,149].

## 3. Environmental Xenobiotics and Their Impact on Epigenetic Regulation

DNA methylation is a well-established epigenetic mechanism that involves the addition of a methyl group to a cytosine (Figure 3) [150]. This process is strongly linked to gene silencing, genomic imprinting, stem-cell specialization, embryonic growth, X-chromosome inactivation, and inflammatory processes [151,152]. It regulates these cellular processes by modulating gene expression and is tightly controlled by specific enzymatic mechanisms [153]. DNA methylation mainly serves to inhibit transcription, but its effects on transcription vary depending on the cellular and genomic environment [154]. Furthermore, abnormalities in this modification are linked to cancer, neurological conditions, atherosclerosis, and various other diseases [155,156,157].

DNA methylation is mainly carried out by a group of proteins called deoxyribonucleic acid methyltransferases (DNMTs), which include DNMT1 (responsible for maintaining methylation patterns during DNA replication), and DNMT3A and DNMT3B (involved in de novo methylation) [158,159]. Although DNMT2 was initially classified as a DNA methyltransferase, it primarily methylates transfer RNA (tRNA) rather than DNA [160]. DNMT3L, on the other hand, lacks catalytic activity but serves as a regulatory co-factor that enhances the activity of DNMT3A and DNMT3B [158,159]. These enzymes facilitate the transfer of a methyl group from S-adenosyl methionine (SAM) to the 5th carbon of cytosine, generating 5-methylcytosine (5mC; Figure 3) [159]. 5mC is particularly abundant in cytosine–phosphate–guanine (CpG) dinucleotides, which are concentrated in CpG Islands found throughout the genome [161]. In mammalian somatic cells, over 98% of genome methylation occurs at CpG dinucleotides [162]. By contrast, in embryonic stem cells (ESCs), up to 25% of DNA methylation occurs at non-CpG sites [163,164]. Moreover, both ESCs and early embryonic (somatic) cells undergo dynamic DNA methylation changes as part of normal epigenetic reprogramming [165].

Some studies have demonstrated that exposure to environmental xenobiotics induces alterations in DNA methylation patterns that are associated with adverse health effects. Table 1 provides an overview of the effects of environmental xenobiotics on DNA methylation signatures and their associated mechanisms.

In addition to DNA methylation, another fundamental layer of epigenetic regulation involves histone modifications (Figure 4). Histones play a crucial role in preserving chromatin organization and are fundamental to the regulation of gene expression [193]. Histone modifications are extensively explored epigenetic alterations, encompassing processes such as acetylation, methylation, phosphorylation, ubiquitination, SUMOylation, adenosine-5′-diphosphate (ADP)-ribosylation, and deamination [194] (Figure 4). These modifications primarily occur in the *N*-terminal tails of histones, influencing chromatin structure by altering its compaction and accessibility. As a result, they play a crucial role in regulating gene transcription and overall chromatin function [195].

Histone acetylation mainly takes place at conserved lysine residues and is regulated by histone acetyltransferases (HATs), which add acetyl groups, and histone deacetylases (HDACs), which remove them [196]. HATs catalyse the transfer of an acetyl group to the ε-amino group of lysine side chains, with acetyl coenzyme A serving as the acetyl donor substrate [196]. In contrast, histone deacetylases (HDACs) remove these acetyl groups, restoring the positive charge of lysine (Figure 4). This deacetylation process helps maintain chromatin structure and stability, contributing to transcriptional repression [197].

Histone methylation mainly takes place on the side chains of lysine (K) and arginine (R) residues (Figure 4). Lysine can be modified by the addition of one, two, or three methyl groups, while arginine can be monomethylated or dimethylated in a symmetric or asymmetric manner [198]. The influence of histone methylation on gene expression is complex, as it can either activate or repress gene transcription depending on its specific location and the degree of methylation [199].

Histone phosphorylation commonly occurs on serine, threonine, and tyrosine residues and plays a crucial role in chromatin remodelling during processes such as DNA damage response, transcription, and mitosis. This modification is typically mediated by specific kinases and reversed by phosphatases [200].

Histone ubiquitination involves the covalent attachment of ubiquitin molecules, usually to lysine residues (Figure 4), and can influence transcriptional activation or repression depending on the histone and site involved. Unlike canonical ubiquitination linked to protein degradation, histone ubiquitination often serves non-proteolytic signalling functions [201]. SUMOylation, the attachment of small ubiquitin-like modifier (SUMO) proteins to lysine residues (Figure 4), is generally associated with transcriptional repression and genome stability, often counteracting the effects of acetylation [202]. ADP-ribosylation of histones, mediated by ADP-ribosyltransferases, introduces ADP-ribose units onto amino acid residues (Figure 4) and is involved in chromatin relaxation and DNA repair [203]. Finally, histone deamination, though less frequently studied, involves the conversion of arginine to citrulline by peptidylarginine deiminases (PADs; Figure 4), which may impact gene regulation and immune responses [204]. Collectively, these diverse post-translational modifications form a complex histone code that orchestrates cellular processes by modulating chromatin accessibility and recruiting effector proteins.

Table 2 summarizes the impact of environmental xenobiotics on histone modifications.

In addition to the epigenetic effects mediated by changes in DNA methylation and histone modifications, ncRNAs represent key modulators of gene activity, especially in the cellular response to environmental agents. Although ncRNAs do not encode proteins, they can interact with other ribonucleic acids (RNAs) and molecules, helping to regulate gene expression, transcription control, splicing, and other important cellular functions [217].

The ncRNA landscape comprises a wide array of RNA molecules, which are generally divided into two main categories: housekeeping ncRNAs and regulatory ncRNAs. Within the regulatory group, the most extensively studied are miRNAs, which are short transcripts ranging from 19 to 25 nucleotides, and lncRNAs, which exceed 200 nucleotides in length. miRNAs typically act as negative regulators of gene expression by binding to target messenger ribonucleic acids (mRNAs) in a sequence-specific manner, leading to their degradation or blocking their translation.

In contrast, lncRNAs can either enhance or suppress gene expression at multiple regulatory levels through interactions with DNA, RNA, or proteins, employing diverse molecular mechanisms [218].

Several environmental xenobiotics have been shown to modulate ncRNA profiles (Table 3), which can influence gene expression, cellular processes, and contribute to the development of various diseases.

In summary, exposure to environmental xenobiotics can trigger profound epigenetic changes, involving mechanisms such as DNA methylation, post-translational histone modifications, and the expression of non-coding RNAs. Understanding these processes offers not only new perspectives for prevention and early diagnosis but also potential opportunities for the development of more targeted epigenetic therapies.

## 4. Xenobiotic-Induced Epigenetic Changes in Disease Pathogenesis

Emerging evidence underscores the pivotal role of epigenetic modifications in the initiation and progression of various diseases, including cancer, neurodegenerative conditions, cardiovascular disorders, and immune dysfunctions [230]. Since several environmental xenobiotics have been consistently associated with epigenetic dysregulation [220,223,231], these mechanisms may play a critical role in the development of chronic diseases, underscoring the need for further research in this field.

### 4.1. Cancer

Environmental xenobiotics can induce profound epigenetic alterations, particularly affecting DNA methylation patterns, histone modifications, and ncRNA expression, which collectively contribute to carcinogenesis (Figure 5). Long-term exposure to heavy metals has been shown to promote the hypomethylation of CpG sites in key oncogenes like nuclear factor kappa B subunit 1 (NFKB1) [169]. In fact, exposure to Cd, whether occupational or environmental, has been associated with an increased risk of cancers, including those of the lung, breast, prostate, pancreas, urinary bladder, and nasopharynx [232]. A likely mechanism involves the upregulation of lncRNA-ENST00000446135, which targets mRNAs involved in cancer [220]. The upregulation of lncRNA-p21, a regulator of cell-cycle progression, by As [219] also suggests a potential mechanism through which As exposure could contribute to dysregulated cell-cycle control and carcinogenesis.

Similarly, tobacco smoke, a complex environmental mixture, has been linked to DNA methylation alterations in genes implicated in lung cancer susceptibility, such as those coding for Kruppel-like factor 6 (KLF6), serine/threonine kinase 32A (STK32A), telomerase reverse transcriptase (TERT), MutS homologue 5 (MSH5), actin alpha 2 (ACTA2), GATA binding protein 3 (GATA3), vesicle transport through interaction with T-SNAREs homologue 1A (VTI1A), and cholinergic receptor nicotinic alpha 5 subunit (CHRNA5) [175]. These epigenetic disruptions are thought to drive the malignant transformation of normal lung cells.

In human bronchial epithelial cells, environmental carcinogens like benzo(*a*)pyrene diol epoxide, a metabolite of tobacco smoke and urban air pollution, increased the expression of DNMTs and caused the hypermethylation-dependent downregulation of cadherin 13 (CDH13) [233]. CDH13 is considered an anti-oncogene, and its downregulation through methylation has been implicated in the initiation and progression of different types of cancer [234,235].

While smoking remains a well-established primary risk factor for lung cancer, a significant proportion of cases have also been associated with air pollution and other environmental contaminants [236,237]. Pollutants such as PM, toxic metals, and nitrogen oxides can infiltrate the respiratory system and potentially trigger epigenetic alterations that contribute to the malignant transformation of normal cells [238].

Ochratoxin A has been shown to increase and decrease methylation in various genes associated with the mammalian target of the rapamycin (mTOR) signalling pathway in rat kidney, which may play a role in the development of kidney cancer [183]. Microcystin-LR also altered DNA methylation, leading to the silencing of tumour suppressor genes, thereby promoting carcinogenesis [239].

Exposure to high levels of aflatoxins also disrupted histone modification processes, inducing epigenetic alterations associated with the development of hepatocellular carcinoma [143,225].

Okadaic acid also shows tumour-promoting effects. In fact, in mammalian cells, it induces DNA hypermethylation [192], and alters histone modifications, affecting DNA damage response and cell-cycle regulation [214]. Moreover, it modulates miRNA expression profiles in gastric adenocarcinoma cells [229]. Altogether, this suggests that the tumour-promoting effects of okadaic acid may arise from epigenetic mechanisms, although oxidative DNA damage and other cytotoxicity-related intermediates may also contribute.

*p*,*p*’-DDT has been shown to impact the expression of miRNAs that upregulate the expression of tumour protein p53 inducible nuclear protein 1 (TP53INP1) and X-linked inhibitor of apoptosis protein (XIAP) genes, which are frequently upregulated in cancers [222].

Other studies have indicated that aflatoxin B1-induced upregulation of miRNA-34a suppressed the Wnt/β-catenin signalling pathway, favouring liver tumorigenesis [225]. Another key miRNA involved in aflatoxin B1-related hepatocellular carcinoma is miRNA-21, which regulates phosphatase and tensin homologue (PTEN) expression and associated signalling pathways, thereby activating the protein kinase B (PKB/AKT) pathway. miRNA-21 is recognised for its role in stimulating cell growth, inhibiting programmed cell death, and facilitating both tissue invasion and the formation of new blood vessels, characteristics that favour tumorigenesis [144,240].

Prolonged exposure to low levels of microcystin-LR upregulated the expression of oncogenic miRNAs [149]. Similarly, altered miRNA profiles were also observed in the livers of mice treated with microcystin-LR, involving miRNAs directly implicated in the development of liver cancer [241].

In parallel, growing evidence suggests that the epigenetic regulation by lncRNAs is a critical factor in modulating gene expression and the progression of diseases [242]. lncRNAs have been increasingly associated with the advancement of pancreatic cancer, and other types of cancer. Evidence indicates that these molecules influence cancer development through multiple regulatory layers, including transcriptional, post-transcriptional, and epigenetic mechanisms. Particularly, aflatoxin B1 has been shown to upregulate the expression of lncRNA-H19, which enhanced the proliferation and invasiveness of hepatocarcinoma cells in vitro [226]. Similarly, other studies have reported an extensive alteration in lncRNA expression in rat liver cells exposed to aflatoxin B1 [243]. Thus, as research in this field progresses, lncRNAs are emerging as promising biomarkers for prognosis and as potential targets for molecular-based therapeutic strategies in oncology [244].

Overall, these findings highlight that environmental exposures to distinct xenobiotics can drive specific epigenetic changes, including hypermethylation of tumour suppressor genes and hypomethylation of oncogenes, thereby promoting the initiation and progression of cancer. Understanding the role of environmental exposures in driving epigenetic changes may open new avenues for prevention and treatment strategies aimed at mitigating the effects of xenobiotics on cancer development.

### 4.2. Neurodegenerative Diseases

Neurodegenerative diseases, including Alzheimer’s (AD) and Parkinson’s (PD) diseases, are characterised by progressive alterations in neuronal function and structure, with multifactorial aetiologies involving complex interactions between genetic and environmental factors. Increased evidence has suggested that environmental factors can induce epigenetic modifications that affect gene expression and contribute to the pathogenesis of these diseases (Figure 5) [245,246].

In AD, DNA methylation alterations have been detected in the hippocampus, particularly in regions involved in neural differentiation, correlating with the accumulation of hyperphosphorylated tau protein, a hallmark of the disease [247]. Recent evidence indicates that exposure to particulate air pollution, particularly metal-rich combustion- and friction-derived nanoparticles, reduced the levels of di- and trimethylated histone H3 at lysine 9 (H3K9me2/me3) in the prefrontal white matter of young individuals living in highly polluted environments [248]. Importantly, these individuals also showed elevated levels of hyperphosphorylated tau and amyloid-β (Aβ) plaques, hallmarks of AD [248]. Although a link between histone hypomethylation and tau hyperphosphorylation or Aβ plaque formation has not been established, this raises the possibility that particulate air pollution-induced histone modifications may increase the risk of AD.

On the other hand, manganese (Mn) exposure has been linked to reduced histone acetylation [207] and diminished expression of the glutamate transporter 1 (GLT-1) [249] and the astrocytic glutamate–aspartate transporter (GLAST) [250], processes that facilitate neurotoxic outcomes. Furthermore, in vitro studies demonstrated that Mn can increase DNA methylation of the Parkin and PTEN-induced putative kinase 1 (PINK1) genes [251], implying that Mn may induce epigenetic alterations in key genes implicated in PD pathogenesis.

Other studies have shown that Pb exposure interferes with long interspersed nuclear element-1 (LINE-1) methylation [252]. Since elevated levels of highly active retrotransposition-competent–long interspersed nuclear element-1 (RC-LINE-1) elements have been connected to an increased risk and progression of PD [253], it is possible that this epigenetic alteration serves as a mechanism through which Pb contributes to PD pathogenesis.

The herbicide paraquat, a known inducer of PD [254], promoted the acetylation of histone H3 in dopaminergic neurons and is linked to a decreased HDAC expression [208]. Moreover, 1-methyl-4-phenyl-1,2,3,6-tetrahydropyridine (MPTP), a neurotoxin extensively used to model PD in animals, particularly in primates and rodents, can increase miRNA-380-3p, contributing to neurotoxicity in dopaminergic neurons [255]. This pathway is considered a likely mechanism underlying MPTP-induced neurodegeneration, although further experimental validation is needed. Additionally, the expression levels of miRNA-34a, miRNA-141, and miRNA-9 were altered in PC12 cells following treatment with the MPTP metabolite 1-methyl-4-phenylpyridinium (MPP^+^) [256].

These findings underscore the critical role of environmental xenobiotics in inducing epigenetic modifications that may influence the onset and progression of neurodegenerative diseases.

### 4.3. Cardiovascular Diseases

Recent research underscores the significant role of environmental xenobiotics in inducing epigenetic changes that contribute to cardiovascular diseases (CVDs; Figure 5). Exposure to toxins such as As, Pb, and fine PM (PM2.5), has been shown to cause alterations in DNA methylation and histone modifications, which can disrupt gene expression related to cardiovascular health.

A study by Jiang et al. [257] demonstrated that participants with higher urinary As levels had more advanced epigenetic ages, indicating accelerated biological ageing. Furthermore, increased As exposure was associated with a higher risk of CVD incidence, mortality related to CVDs, and all-cause mortality, with approximately 21% to 23% of the CVD incidence risk attributed to accelerated epigenetic ageing [257].

In a related study, Domingo-Relloso et al. [258] investigated whether differentially methylated positions (DMPs) in blood DNA could mediate the relationship between As exposure and CVD incidence and mortality. They identified 20 DMPs as potential mediators of CVD incidence and 13 DMPs associated with CVD mortality. Many of these DMPs were annotated to genes involved in diabetes, suggesting that diabetes may play a significant role in the cardiovascular risks induced by As exposure [258].

PM2.5 is another environmental pollutant, with a well-established correlation with negative human health outcomes, particularly CVDs [259,260]. Studies have shown that PM2.5 can induce the production of reactive oxygen species (ROS) and nitric oxide (NO) in cardiovascular endothelial cells, contributing to oxidative stress [261]. This oxidative stress can lead to epigenetic modifications, potentially linking oxidative damage to the development of atherosclerosis [262]. Other studies have highlighted a strong association between epigenetic regulation and cardiovascular risks following PM2.5 exposure, with mechanisms involving histone acetylation, interferon γ methylation, DNA methylation, m6A RNA methylation, ncRNAs, histone modifications, and chromatin remodelling [263,264]. Notably, m6A RNA methylation has been identified as a dynamic process influenced by PM2.5 exposure, affecting gene transcription. Thus, m6A methylation is emerging as a potential bridge linking PM2.5 exposure to CVDs [263,264].

Collectively, these findings highlight the critical role of environmental xenobiotics in driving epigenetic changes that accelerate CVD risk.

### 4.4. Immune Disorders

Environmental xenobiotics, including industrial pollutants and heavy metals, can also disrupt immune tolerance by inducing epigenetic modifications (DNA methylation and histone modifications). Such modifications may lead to the development of autoimmune diseases by affecting gene expression in immune cells, especially during critical developmental windows (Figure 5).

Epigenetic changes in immune-related genes, such as those seen in systemic lupus erythematosus (SLE), can interfere with immune regulation, triggering autoantibody production and promoting tissue destruction. [265,266]. DNA hypomethylation, for example, is thought to occur primarily due to a reduced expression or activity of DNMTs [267], which has been linked to the onset of lupus-like diseases in experimental models [268]. Exposure to environmental pollutants like cigarette smoke and heavy metals further exacerbated these changes, enhancing oxidative stress and inflammatory gene expression [269].

Other studies have revealed significant methylation changes in genes associated with various cellular processes, including immune and inflammatory responses. In a study by Stepanyan [169], prolonged exposure to environmental metals resulted in the hypomethylation of CpG sites in the NFKB1 gene, which is crucial for immune response regulation [270].

In parallel, miRNA-146a regulates various immune system processes, including inflammation and the response to infection. Its downregulation by a mixture of pesticides (chlormequat chloride, pirimiphos-methyl, glyphosate, tebuconazole, chlorpyrifos-methyl, and deltamethrin) [221] suggests that these environmental xenobiotics may disrupt the immune system.

These findings underscore the critical role of environmental xenobiotics in disrupting immune tolerance through epigenetic modifications, highlighting their potential contribution for the development of autoimmune diseases.

## 5. Critical Windows of Susceptibility

The epigenome undergoes dynamic reprogramming during embryonic development, establishing critical gene-expression patterns for cellular differentiation and organogenesis [271]. This stage, as well as the neonatal and infant stages of development, encompassing approximately the first 1000 days post-conception, are periods of rapid cell division and heightened epigenetic remodelling, representing critical windows of susceptibility to xenobiotic-induced epigenetic dysregulation (Figure 6A). Given the high epigenomic plasticity and sensitivity to environmental cues during these developmental periods, even transient exposures can yield long-lasting effects [272]. The association between intrauterine and neonatal exposure to environmental factors and the development of adult diseases has been hypothesized in several studies that date back to the 1930s, though it was more definitively established by Barker and colleagues in the 1990s [273,274,275,276], with the “foetal basis of adult disease” hypothesis (see review [277] for historical background). Since then, the effects of all kinds of environmental xenobiotic exposure during these critical developmental stages have been an object of investigation in different species.

Maternal exposure during gestation directly shapes the foetal epigenome via the placental transfer of xenobiotics, altering DNA methylation and histone patterns in developing tissues and affecting the offspring phenotypes into adulthood (Figure 6B).

Studies in rodents established this direct link using the viable yellow agouti (*A^vy^*) mouse model. For instance, prenatal exposure to BPA has been shown to alter DNA methylation patterns, shifting offspring coat colour toward yellow, representing clear evidence of a phenotype alteration due to epigenetic modifications [278]. Similarly, in utero exposure to genistein (a soy-derived phytoestrogen) at doses comparable to those consumed by humans was associated with DNA hypermethylation, leading to a shift in the offspring coat colour of *A^vy^* mouse towards brown (pseduoagouti), and was correlated with a reduced risk for their obesity [279]. In humans, maternal exposure to environmental pollutants such as heavy metals, EDCs, air pollutants, and POPs during pregnancy has also been linked to altered epigenetic patterns and, consequently, to different health outcomes in the children (reviewed in [280,281,282]). One specific example is maternal smoking, which represents a source of PAHs and nicotine, and has been shown to induce DNA methylation changes in genes linked to respiratory diseases in the exposed children, with epigenetic alterations persisting at least until late adolescence [283,284,285,286,287]. While DNA methylation is the most readily accessible biomarker for detecting epigenetic alterations, research also indicates that prenatal xenobiotic exposure can lead to post-translational modifications of histones and changes in ncRNAs [280].

Beyond intrauterine exposures during pregnancy, studies in neonates and infants have also linked environmental xenobiotic exposure to epigenetic alterations that are associated with increased risks for diseases later in life (Figure 6B) [288]. Additionally, paternal xenobiotic exposure also impacts offspring epigenetic patterns, although in an indirect manner (see the section on “Transgenerational epigenetic effects of xenobiotics”).

The increased susceptibility to epigenetic dysregulation during early life stages stems from several physiological and behavioural aspects: the rapid cell division, incomplete DNA repair mechanisms, and ongoing epigenetic reprogramming are inherent aspects to embryogenesis and postnatal growth, where minor errors can have serious impacts on the child’s development and health outcomes [289,290,291]; the foetus and newborns solely obtain nutrients from the mother’s sources, being exposed to persistent pollutants in the mothers’ food chain that pass through the physiological barriers [292]; children ingest more food and water per unit of body weight, resulting in higher concentrations of xenobiotics in their bodies compared to adults [293,294]; the metabolism of embryos, newborns, and children is considered immature in terms of detoxification capacity, leading to either reduced or prolonged exposure to xenobiotics within their bodies, depending on the life stage and the xenobiotic [295,296]; infants are particularly prone to increased exposure to environmental pollutants through the ingestion and dermal absorption of contaminated dust particles while playing on the ground [297,298]. All these aspects have the potential to impact epigenetic plasticity in such critical developmental periods, resulting in molecular changes that manifest as phenotypic alterations over time [288,299].

## 6. Transgenerational Epigenetic Effects of Xenobiotics

Increasing evidence suggests that xenobiotic-induced epigenetic modifications can persist across multiple generations, a phenomenon known as transgenerational epigenetic inheritance (Figure 7A,B) [300]. This refers to the transmission of epigenetic marks to descendants beyond the directly exposed generation, with potential long-term consequences for population health, contributing to disease susceptibility over time. In mammals, true transgenerational effects (observed in completely unexposed generations) require the study of the xenobiotic-induced epigenetic alterations in the third generation of descendants (F3 generation) from an exposed ancestor (F0 generation) (Figure 7A). This is because when exposure occurs during pregnancy, F1-generation embryos and F2-generation germ cells are directly exposed during F0-generation pregnancy [301]. For mammalian paternal exposure or in organisms with external embryonic development, such as zebrafish, transgenerational effects can be studied in the F2 generation [302].

Several studies provide strong evidence of this phenomenon in response to environmental xenobiotic exposure, with effects varying depending on the specific compound, the timing of exposure (e.g., prenatal vs. adult), and the species studied. One significant study exposed pregnant rats (F0 generation) to the endocrine disruptors vinclozolin, a fungicide, or methoxychlor, a pesticide. For both compounds’ exposure, the F1 generation exhibited altered DNA methylation patterns in the sperm that persisted into the F4 generation, accompanied by reproductive defects (Figure 7C) [303]. Exposure to BPA in zebrafish and mice altered histone modifications and gene-expression patterns in reproductive organs, the heart, and the brain that persisted into the next generations (F2 in zebrafish and F3 in mice) (Figure 7C) [304,305,306,307,308]. Similarly, paternal exposure to microcystin-LR led to decreased hatching success, slower heart rates, and lower body weights in both the F1 and F2 generations of zebrafish [142]. Moreover, in *C. elegans*, exposure to heavy metals (e.g., Cd), NPs, and organic pollutants induced epigenetic changes that resulted in reproductive, behavioural, and metabolic changes transmissible for over four generations (Figure 7C) [309,310,311,312,313,314,315].

While direct transgenerational studies in humans are challenging due to ethical constraints and long generation times, some data indicate that humans are also susceptible to this effect. Prenatal exposure to diethylstilboestrol, a synthetic oestrogen and EDC, has been associated with increased risk of reproductive defects in grandchildren, suggesting epigenetic transmission (Figure 7C) [316]. Additionally, recent studies suggested that the descendants (F1 to F3) of males exposed to tobacco smoke before puberty show increased body fat during childhood, adolescence, and early adulthood (Figure 7C) [317,318]. As this is among the first evidence of xenobiotic-induced transgenerational effects beyond the second generation, it raises the possibility of epigenetic involvement, as the original idea that genetic mutations can fully explain heritability is no longer accepted [319,320]. Moreover, given that obesity is widely accepted to result from a complex interplay of genetic, epigenetic, and environmental influences [321], it is likely that tobacco smoke-induced epigenetic changes may contribute to this outcome, although further investigation is required.

The persistence of epigenetic changes across generations implies that these marks escape the natural epigenetic reprogramming events that occur during gametogenesis and early embryonic development. The proposed mechanisms often involve disrupted DNA methylation patterns transmitted to the subsequent generations via the germline [303,322], histone modifications [304,308], or ncRNAs that alter miRNAs profiles in the germline, which persist in the following generations [323]. The environmental persistence of some xenobiotics may lead to low-dose exposures that also cause epigenetic load accumulation, where successive generations inherit combined epigenetic disruptions from multiple ancestors. Further studies are necessary to clarify the underlying pathways by which these changes may escape erasure during gamete formation.

The transgenerational epigenetic effects of environmental xenobiotics may pose significant challenges for population health and evolution overtime. These effects are linked to a range of disorders, including infertility, obesity, cancer, and neurodevelopmental conditions in animal models [300]. If similar effects occur in humans, populations exposed to pollutants decades ago could exhibit increasing disease burdens today. Additionally, persistent epigenetic changes may alter phenotypic characteristics, potentially changing populations traits over time, as an adaptation to certain pollutants or, conversely, higher sensitivity to their effects, resulting in disease susceptibility. The long-term monitoring of environmental xenobiotic-exposed populations and their descendants is essential for assessing the real impact of those compounds on human health, as current safety assessments often focus on acute or chronic toxicity in exposed individuals, potentially underestimating multigenerational toxicity. Analyses of epigenetic biomarkers in sperm, such as differential methylation at obesity-related genes, could enable the early detection of population-level risks.

## 7. Reversibility of Xenobiotic-Induced Epigenetic Changes

Unlike genetic mutations, epigenetic modifications are generally considered reversible, as they result from dynamic mechanisms such as DNA methylation and histone modifications [324]. Several studies have demonstrated that xenobiotic-induced epigenetic alterations can, under specific conditions, be reversed, offering promising alternatives for therapeutic intervention.

One study explored the role of 8-oxoguanine deoxyribonucleic acid glycosylase 1 (OGG1) and the sex-specific effects of trichostatin A (TSA), a potent and reversible HDAC inhibitor, in mitigating neurodevelopmental impairments caused by prenatal ethanol exposure [325]. TSA treatment alone was found to influence neurodevelopment and partially reverse ethanol-induced alterations in an OGG1- and sex-dependent manner. Furthermore, TSA improved performance in learning, memory, and cognitive function tasks, supporting the hypothesis that histone acetylation and the resulting upregulation of gene expression are critical for memory formation. TSA also reversed ethanol-induced behavioural alterations such as tolerance, anxiety, and ethanol consumption, primarily through HDAC inhibition and increased histone acetylation [325]. These findings suggest that HDAC inhibition can partially reverse ethanol-induced neurobehavioral alterations, highlighting the involvement of epigenetic mechanisms in these outcomes, rather than proving the reversibility of specific epigenetic changes.

A study investigating the effects of cigarette smoke exposure on bone repair-related genes also reported that the administration of resveratrol, a natural polyphenol with known epigenetic activity [326,327], reversed both epigenetic and transcriptional changes induced by cigarette smoke in rats. Specifically, resveratrol enhanced the transcription of Dnmt3a, Dnmt3b, and sirtuin 1 (Sirt1), while reducing the expression of the receptor activator of nuclear factor κB ligand (Rankl) and tartrate-resistant acid phosphatase (Trap), genes associated with bone resorption [328]. These findings suggest that epigenetic alterations induced by environmental toxins can be pharmacologically reversed, supporting the potential for therapeutic interventions in xenobiotic-induced pathologies.

Clear indications of reversibility of specific epigenetic changes were provided in a longitudinal human study conducted by Richmond et al. [329]. They showed that maternal smoking during pregnancy was associated with altered DNA methylation in offspring at birth, and at ages 7 and 17, in a cohort from the Avon Longitudinal Study of Parents and Children (ALSPAC). While some methylation changes related to development and metabolism persisted into adolescence, others were reversible over time, suggesting a gene- and site-specific pattern of epigenetic plasticity [329].

Further insights into the reversible nature of xenobiotic-induced epigenetic changes were provided by studies on dimethyl sulfoxide (DMSO) exposure [330]. Within 12 to 24 h of exposure, increased expression of Tet and growth arrest and DNA damage-inducible 45 (Gadd45) genes (key regulators of DNA hydroxymethylation and nucleotide excision repair, respectively [331,332,333]) was observed, along with decreased expression of DNA methylation-related genes such as Dnmt1, Dnmt3b, and helicase, lymphoid specific (Hells) [4]. However, by day five, the epigenetic effects of DMSO, including changes in global and promoter-specific methylation and hydroxylation, were diminished or fully reversed [330].

Collectively, these studies underscore the inherent reversibility of many epigenetic changes induced by environmental xenobiotics. Under certain conditions, changes in lifestyle, or the cessation of exposure, the epigenome may return to a baseline state. Recognizing and controlling this flexibility is essential for developing effective strategies to mitigate the long-term health risks associated with environmental xenobiotic exposure.

## 8. Conclusions

Environmental xenobiotics, including heavy metals, EDCs, pesticides, and air pollutants, represent significant problems to human health by altering gene expression through epigenetic mechanisms. These include DNA methylation changes, a variety of histone modifications, and the altered expression of ncRNAs, each of which plays a critical role in regulating fundamental cellular processes such as gene transcription, and cell-cycle control. Consequently, disruptions in these tightly controlled mechanisms can potentially contribute to the onset and progression of a broad spectrum of diseases, including cancer, neurodegenerative diseases, CVDs, and dysfunction of the immune system.

Importantly, exposure during sensitive developmental windows, including embryogenesis and shortly after birth, has been shown to induce long-lasting epigenetic changes. These alterations may persist throughout an individual’s lifetime, potentially influencing health outcomes and increasing susceptibility to chronic diseases. Moreover, evidence from animal models and limited, but increasing amounts of human data reveal that environmental xenobiotic-induced epigenetic changes can be inherited across multiple generations. This phenomenon, known as transgenerational epigenetic inheritance, suggests that the effects of environmental xenobiotics may extend beyond the directly exposed individual, presenting long-term risks to population health.

At the same time, the reversibility of many epigenetic changes offers a promising opportunity for intervention. Studies have demonstrated that pharmacological agents, such as histone deacetylase inhibitors, and naturally occurring compounds like resveratrol can reverse xenobiotic-induced epigenetic alterations. Additionally, reductions in exposure or lifestyle modifications may also promote the repair of normal epigenetic patterns. This plasticity of the epigenome opens new opportunities for therapeutic development and early prevention strategies.

Overall, these findings highlight the urgent need for a different perspective on how environmental health risks are assessed. Traditional toxicological assessments, which primarily focus on acute or short-term effects in directly exposed individuals, may significantly underestimate the true consequences of environmental xenobiotics in human health. Thus, a more comprehensive, multigenerational approach is needed, accounting for the long-term, heritable, and potentially reversible nature of epigenetic alterations.

## 9. Future Perspectives

The investigation into how environmental xenobiotics modulate epigenetic mechanisms and influence human health has increased exponentially over the past few years. These modifications can have strong consequences on human health and contribute to a broad spectrum of pathologies, such as cancer and neurodegenerative diseases [7,8]. Despite this increasing evidence, which has guided public health policies, the full impact of environmental xenobiotics on human health through epigenetic mechanisms still requires further investigation.

One of the most critical current limitations is the scarcity of longitudinal studies tracking the long-term effects of environmental xenobiotics on epigenetic markers and their implications for human health [334]. While cross-sectional studies have provided valuable insights into the association between exposure to environmental xenobiotics and epigenetic changes [335,336,337], they do not elucidate the dynamic nature of epigenetic modifications over time. Thus, to fully understand the impact of environmental xenobiotics on epigenetic signatures, future research must focus on prospective longitudinal studies that follow individuals or populations over time. These studies will help identify the timing and duration of exposure to environmental xenobiotics that may lead to epigenetic changes that last for long periods. Additionally, such studies will help characterise whether epigenetic changes caused by environmental xenobiotics are reversible or permanent.

The transgenerational effects of environmental xenobiotics represent another important research question. Many epigenetic modifications can be transmitted to subsequent generations, potentially influencing the health of descendants even in the absence of direct exposure [338]. This phenomenon, known as epigenetic inheritance, has been observed in animal models [303,339,340,341,342], but human studies are fairly limited. Thus, it is crucial to understand how the exposure to environmental xenobiotics, particularly during prenatal or early life, which represent critical developmental periods, can lead to transgenerational epigenetic changes.

Another area that requires detailed investigation is the impact of environmental xenobiotics on the epigenetic background of vulnerable populations, which include children, the elderly, pregnant women, and individuals with comorbidities. Particularly for pregnant women, alterations in the foetal epigenome resulting from prenatal exposure to environmental xenobiotics can result not only in immediate health problems but can also increase susceptibility to diseases later in life. Such knowledge may guide the development of new regulatory guidelines and public health policies that address these particular populations.

On the other hand, since the relationship between environmental xenobiotics, epigenetic modifications, and human health is complex, a more integrated research approach becomes necessary. At this level, multi-omics strategies combining genomics, transcriptomics, epigenomics, metabolomics, and proteomics can help us to understand how exposure to environmental xenobiotics influences health at a molecular level. These strategies have the potential to identify biomarkers of exposure and susceptibility, increasing our understanding of disease mechanisms and possibly predicting outcomes. For example, integrating transcriptomic data, which provides information about changes in gene expression [343], with epigenomic information can potentially reveal direct relationships between exposure to environmental xenobiotics and abnormal gene expression, contributing to diseases involving epigenetic mechanisms.

While identifying the epigenetic changes induced by environmental xenobiotics is important, understanding the precise biological mechanisms behind these modifications is even more crucial. Moreover, it is essential to understand how epigenetic modifications triggered by environmental xenobiotics lead to long-term effects, resulting in diseases such as cancer or neurodegeneration. At this level, advanced technologies such as clustered regularly interspaced short palindromic repeats (CRISPR)-Cas-based epigenome editing and single-cell sequencing show tremendous potential for elucidating the dynamic nature of these epigenetic modifications at both the individual cell level and within complex tissues. Additionally, future research should explore how these epigenetic changes interact with other cellular processes, such as inflammation or oxidative stress, both frequently associated with environmental xenobiotics [344], to promote disease.

Overall, answering these research questions will likely enhance our understanding of the true role that epigenetic mechanisms play in mediating the harmful effects of environmental xenobiotics on human health.

## Figures and Tables

**Figure 1 jox-15-00118-f001:**
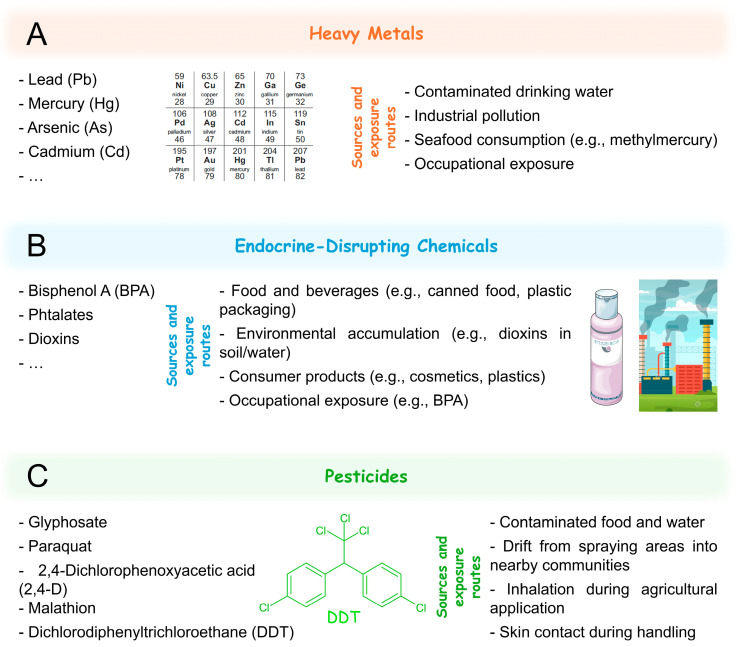
Sources and exposure routes of (**A**) heavy metals, (**B**) endocrine-disrupting chemicals, and (**C**) pesticides. For each group, key examples are listed alongside common routes of human exposure, such as contaminated food, water, air, occupational contact, and consumer products. The illustration incorporates elements from Servier Medical Art, and the chemical structure was generated using the ChemSketch freeware (version 2023.2.4; ACD/Labs, Toronto, ON, Canada).

**Figure 2 jox-15-00118-f002:**
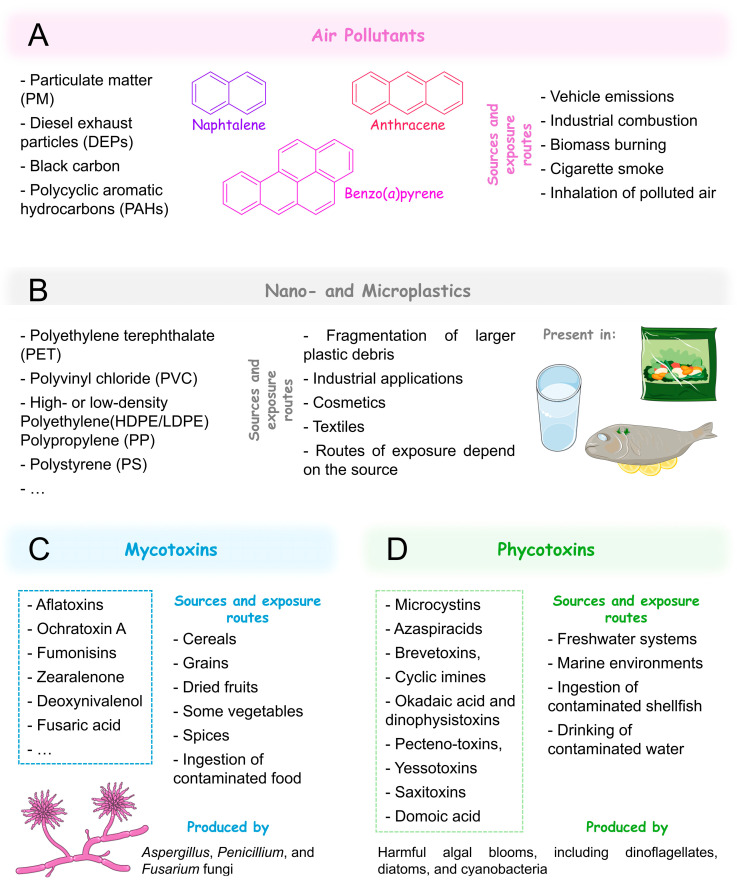
Sources and exposure routes of (**A**) air pollutants, (**B**) nano- and microplastics, (**C**) mycotoxins, and (**D**) phycotoxins. For each group, key examples are listed alongside common routes of human exposure, such as contaminated food, water, air, occupational contact, and consumer products. The illustration incorporates elements from Servier Medical Art, and the chemical structures were generated using the ChemSketch freeware (version 2023.2.4; ACD/Labs, Toronto, ON, Canada).

**Figure 3 jox-15-00118-f003:**
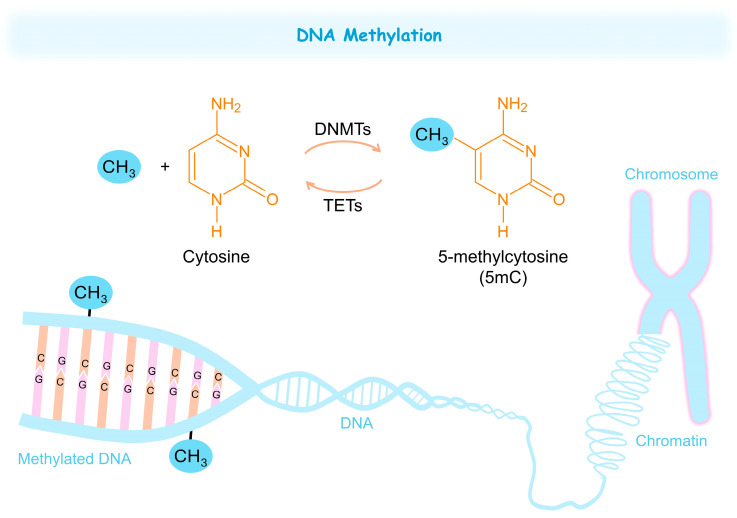
DNA methylation process. Deoxyribonucleic acid methyltransferases (DNMTs) catalyse the addition of a methyl group (CH_3_) to cytosine, forming 5-methylcytosine (5mC), predominantly abundant in cytosine–phosphate–guanine (CpG) sites. Conversely, ten-eleven translocation (TET) enzymes mediate the removal of the methyl group, returning 5-methylcytosine to unmethylated cytosine. The illustration incorporates elements from Servier Medical Art, and the chemical structures were generated using the ChemSketch freeware (version 2023.2.4; ACD/Labs, Toronto, ON, Canada).

**Figure 4 jox-15-00118-f004:**
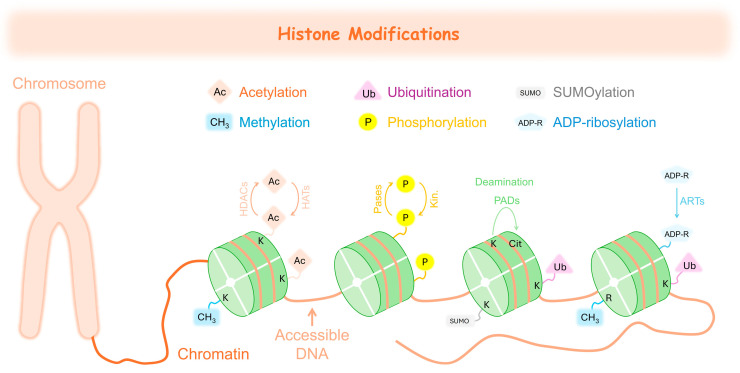
An overview of histone modifications. Histone post-translational modifications primarily occur on the *N*-terminal tails of histones and include acetylation, methylation, phosphorylation, ubiquitination, SUMOylation, adenosine 5′-diphosphate (ADP)-ribosylation, and deamination. Histone acetylation involves the addition of acetyl groups (Ac) to lysine residues (K) and is catalysed by histone acetyltransferases (HATs). Histone deacetylases (HDACs) are responsible for removing acetyl groups (Ac) from histones. Histone methylation corresponds to the addition of methyl groups (CH_3_) to lysine (K) or arginine (R) residues. Alternatively, histone phosphorylation (P) can occur on serine, threonine, and tyrosine residues. Ubiquitination attaches ubiquitin molecules (Ub) to histones, usually on lysine residues (K), while SUMOylation involves the addition of SUMO proteins (SUMO) to lysine residues (K). ADP-ribosylation, catalysed by ADP-ribosyltransferases (ARTs), introduces ADP-ribose units (ADP-R) onto amino acid residues. Finally, deamination involves the enzymatic conversion of arginine (R) into citrulline (Cit) by peptidylarginine deiminases (PADs). Altogether, these modifications regulate the accessibility of DNA to transcriptional machinery and play essential roles in epigenetic regulation. The illustration incorporates elements from Servier Medical Art.

**Figure 5 jox-15-00118-f005:**
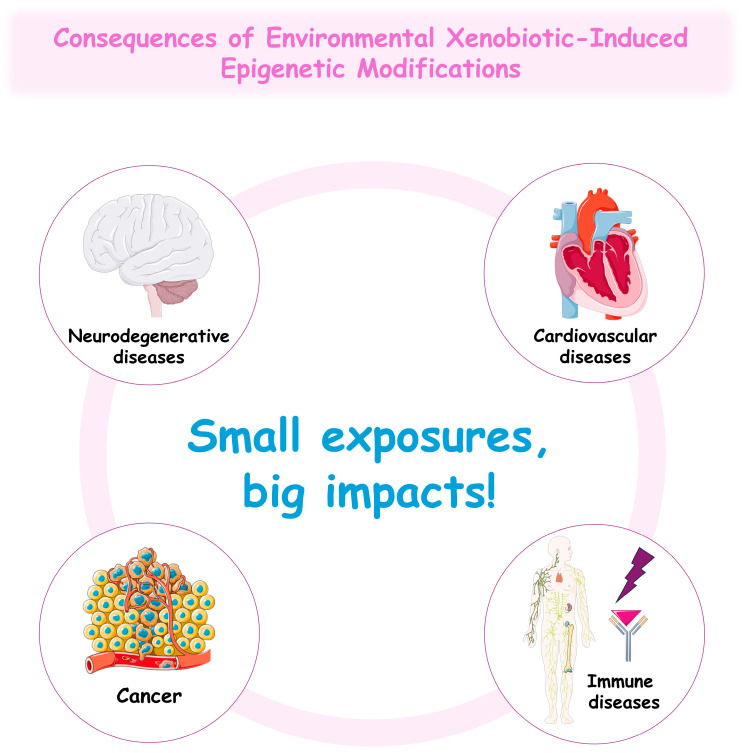
The consequences of environmental xenobiotic-induced epigenetic modifications. Even low-level or early exposures to environmental xenobiotics can lead to epigenetic effects that result in a range of adverse health outcomes. These effects can contribute to the global incidence of chronic diseases. The illustration incorporates elements from Servier Medical Art.

**Figure 6 jox-15-00118-f006:**
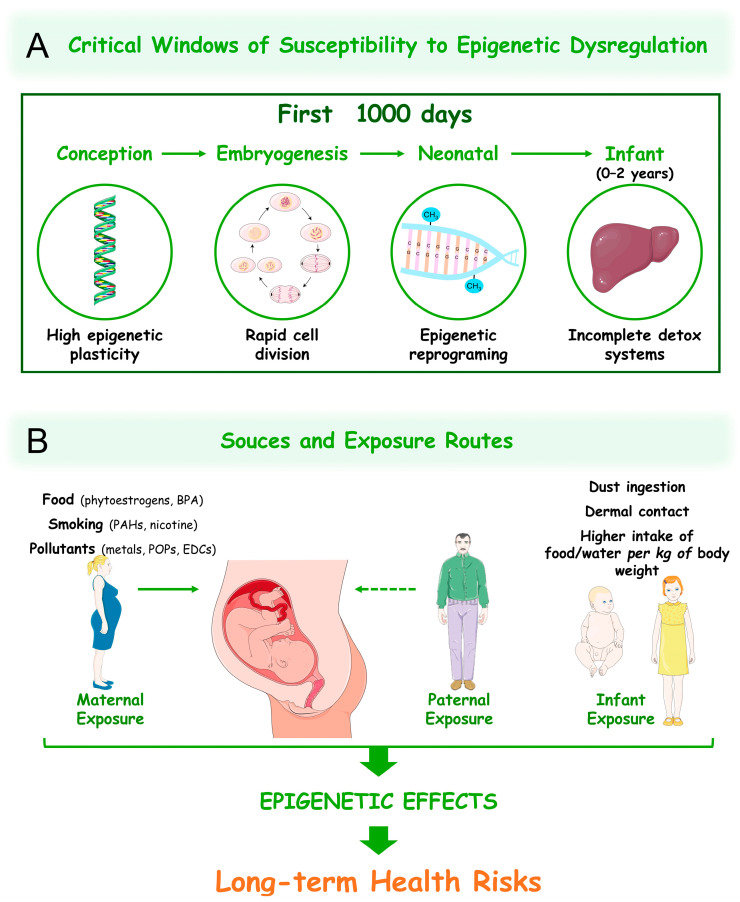
Critical windows of susceptibility to epigenetic dysregulation by environmental xenobiotics. (**A**) The first 1000 days of life (from conception to two years of age) represent a highly sensitive period characterised by high epigenetic plasticity, rapid cell division, epigenetic reprogramming, and immature detoxification systems. (**B**) Major sources and routes of xenobiotic exposure include food, smoking, and environmental pollutants, affecting both maternal and paternal contributions. Infants are also exposed through dust ingestion, dermal contact, and relatively higher intake of food and water per kg of body weight. These exposures can lead to epigenetic alterations associated with long-term health risks. The illustration incorporates elements from Servier Medical Art.

**Figure 7 jox-15-00118-f007:**
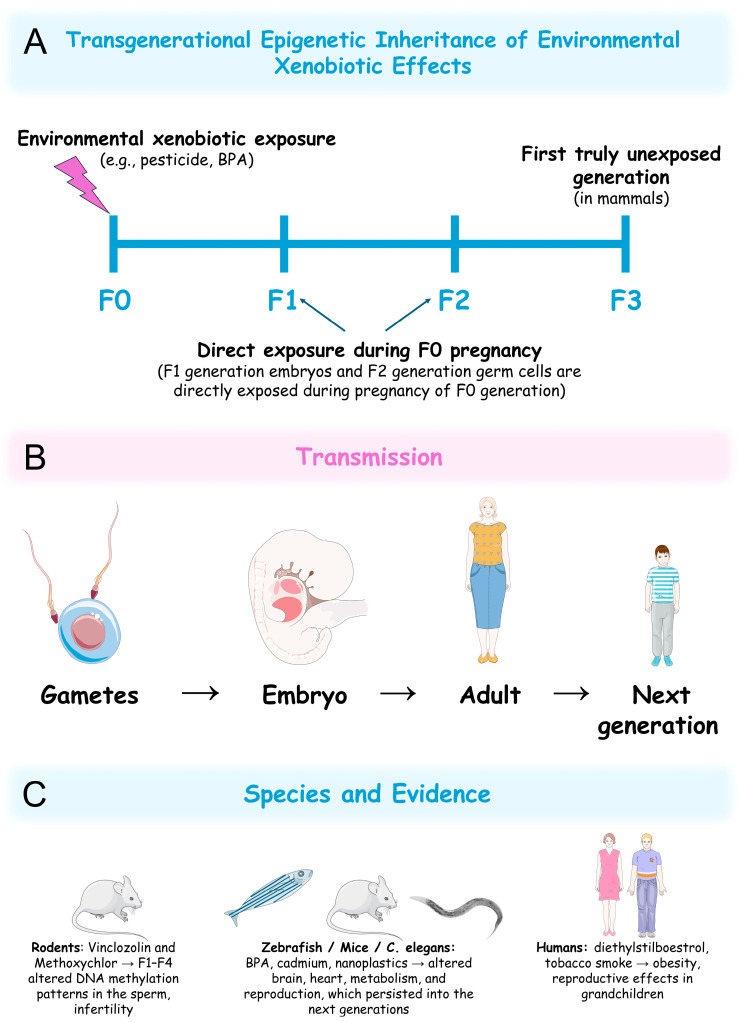
The transgenerational epigenetic inheritance of xenobiotic effects: mechanisms and evidence. (**A**) The exposure of the F0 (parental) generation to environmental xenobiotics during pregnancy (e.g., pesticides, BPA) can affect not only the F1 generation (developing embryo) but also the F2 generation (germ cells within the embryo), making the F3 generation the first to be truly unexposed in mammals. (**B**) Epigenetic marks altered by xenobiotics can be transmitted through the germline, affecting the embryo, adult organism, and potentially the next generation. (**C**) Experimental evidence from multiple species supports transgenerational epigenetic inheritance. The illustration incorporates elements from Servier Medical Art.

**Table 1 jox-15-00118-t001:** Impact of environmental xenobiotics on DNA methylation.

Xenobiotic	Effect on DNA Methylation	Mechanism	References
**Cadmium (acute exposure—24 h to 1 week)**	Decreased DNA methylation	Inhibition of DNMTs	[166]
**Cadmium (prolonged exposure—10 weeks)**	Increased DNA methylation	Increased activity of DNMTs	[166]
**Heavy metals (e.g., arsenic, lead, aluminium)**	DNA methylation alterations, especially in promoter regions	Interference with DNMTs	[167,168]
**Heavy metals**	Hypomethylation of NFKB1 gene	-----	[169]
**Dioxins**	Hypomethylation and alterations in gene expression	Interaction with aryl hydrocarbon receptor, affecting DNMT expression	[170,171]
**Phthalates**	Alterations in DNA methylation, particularly in reproductive tissues	Disruption of epigenetic signalling and hormonal pathways	[172,173]
**Bisphenol A**	Alterations in methylation of genes involved in hormonal function	Interaction with hormonal receptors and DNMT activity	[172,174]
**Tobacco**	DNA hypomethylation	-----	[175]
**Benzo(*a*)pyrene**	DNA hypomethylation	DNMTs inhibition	[176]
**Benzo(*a*)pyrene diol epoxide**	DNA hypermethylation	Recruitment of DNMT3A	[177]
**Polystyrene nanoplastics**	DNA hypomethylation/hypermethylation	-----	[178]
**Aflatoxin B1**	Hypermethylation of p21 promotor	-----	[179]
	DNA hypermethylation	-----	[180,181]
	DNA hypomethylation	-----	[180]
	Decreased LINE-1 and Sat2 promotor methylation	-----	[148]
**Ochratoxin A**	Global DNA hypermethylation	----	[182]
	Global DNA hypermethylation	Increased DNMT1, and DNMT3B expression, but decreased DNMT3A expression	[146]
	DNA hypermethylation (Tbc1d5, Arap2, Ano6, Cul2, and Dlg2 gene promoters)	-----	[183]
	DNA hypomethylation (Cpne4, Pdpk1, Spop, Ogdh, Dock3, and Rptor gene promoters)	-----	
	DNA hypomethylation	-----	[184]
**Mixture of aflatoxin, zearalenone, and deoxynivalenol**	DNA hypermethylation	-----	[185]
**Zearalenone**	Global DNA hypermethylation	Increased DNMT1 expression	[186]
		-----	[145]
	Global DNA hypomethylation	-----	[187]
**Zearalenone, fumonisin B1, and deoxynivalenol, individually or in a mixture**	Global DNA hypermethylation	-----	[188]
**Deoxynivalenol**	Global DNA hypermethylation	Increased DNMT1 and DNMT3B expression	[189]
**Fusaric acid**	DNA hypomethylation	Decrease DNMT1, DNMT3A, and DNMT3B expression, and increase MBD2 expression	[190]
**Microcystin-LR**	DNA hypermethylation	Increased DNMT3A and DNMT3B expression	[191]
	Global DNA hypermethylation and increased DNA methylation of bdnf gene promoter	Increased expression of DNMTs	[142]
	DNA hypomethylation (dio3 and gad1 gene promoters) in F1 generation	-----	[142]
**Okadaic acid**	DNA hypermethylation	-----	[192]

Ano6, anoctamin 6; Arap2, ankyrin repeat and PH domain 2; bdnf, brain-derived neurotrophic factor; Cpne4, copine 4; Cul2, cullin-2; dio3, deiodinase, iodothyronine type III; Dlg2, discs large MAGUK scaffold protein 2; DNA, deoxyribonucleic acid; DNMT1, DNA methyltransferase 1; DNMT3A, DNA methyltransferase 3A; DNMT3B, DNA methyltransferase 3B; DNMTs, DNA methyltransferases; Dock3, dedicator of cytokinesis 3; LINE-1, long interspersed nuclear element-1; MBD2, methyl-CpG binding domain protein 2; NFKB1, nuclear factor kappa B subunit 1 gene; Ogdh, oxoglutarate dehydrogenase; p21, cyclin-dependent kinase inhibitor 1A; Pdpk1, 3-phosphoinositide-dependent protein kinase-1; Rptor, regulatory associated protein of mTOR complex 1; Sat2, satellite 2 DNA; Spop, speckle-type POZ protein; Tbc1d5, TBC1 domain family member 5.

**Table 2 jox-15-00118-t002:** Impact of environmental xenobiotics on histone modifications.

Xenobiotic(s)	Effects on Histones	Mechanism	References
**Cadmium**	Increased H3K4me3 and H3K9me2 levels	Reduced activities of H3K4 and H3K9 demethylases; no changes on KDM5A and KDM3A	[205]
**Hexavalent chromium [Cr(VI)]**	Increased levels of H3K9me2, H3K9me3, H3K4me2, and H3K4me3	Increased expression of G9a histone methyltransferase	[206]
	Decreased levels of H3K27me3, and H3R2me2	-----	
**Manganese**	H3K27 hypoacetylation	Increased expression of HDAC3	[207]
**Paraquat**	Acetylation of histone H3	Decreased HDAC expression	[208]
**Dieldrin**	Histones H3 and H4 hyperacetylation	Reduced proteasomal degradation of HATs	[209]
**Cigarette smoke**	Increased acetylation and methylation of histone H3 and histone H4	Alteration of HATs, HDACs, and DNMT activity	[210]
	Decreased histone acetylation	Reduced HDAC2 activity	[211]
**Aflatoxin B1**	Increased levels of H3K9me3	-----	[181]
	Decreased levels of H3K27me3 and H3K4me2	-----	
	Increased levels of H3K27me3 and H2AK119Ub	-----	[147]
**Ochratoxin A**	Decreased acetylation at H3K9 and H3K14	Inhibition of HATs	[212]
**Mixture of aflatoxin, zearalenone, and deoxynivalenol**	Increased levels of H3K9me3 and H4K20me3, and decreased levels of H3K27me3 and H4K20me2	-----	[185]
**Zearalenone**	Increased levels of H3K9me3, H3K9ac, and H3K27me3	Increased expression of HAT1, KAT2B, ESCO1, PRMT6, and SETD8	[145]
**Microcystin-LR**	Decreased levels of H3K4me2, H3K4me3, and H3K36me3	Increased expression of KDM5B	[213]
**Okadaic acid**	Increased phosphorylation of histones H1 and H3	Likely inhibition of phosphatases 1 and 2A	[214]
	Hyperphosphorylation of histone H3	Likely inhibition of phosphatases 1 and 2A	[215]
	Increased levels of H4K5ac and altered spatial distribution of H3S10ph	-----	[216]

DNMTs, DNA methyltransferases; ESCO1, establishment of cohesion 1 homologue 1; G9a, euchromatic histone–lysine N-methyltransferase 2 (EHMT2); H2AK119Ub, monoubiquitinated histone H2A at lysine 119; H3K4me2, dimethylated of histone H3 at lysine 4; H3K4me3, trimethylated histone H3 at lysine 4; H3K9, lysine 9 of histone H3; H3K9ac, acetylated histone H3 at lysine 9; H3K9me2, dimethylated histone H3 at lysine 9; H3K9me3; trimethylated histone H3 at lysine 9; H3K14, lysine 14 of histone H3; H3K27, lysine 27 of histone H3; H3K27me3, trimethylated histone H3 at lysine 27; H3K36me3, trimethylated histone H3 at lysine 36; H3R2me2, dimethylated histone H3 at arginine 2; H3S10ph, phosphorylated histone H3 at serine 10; H4K5ac, acetylated histone H4 at lysine 5; H4K20me2, dimethylated histone H4 at lysine 20; H4K20me3, trimethylated histone H4 at lysine 20; HAT1, histone acetyltransferase 1; HATs, histone acetyltransferases; HDAC2, histone deacetylase 2; HDAC3, histone deacetylase 3; HDACs, histone deacetylases; KAT2B, lysine acetyltransferase 2B; KDM3A, histone H3K9 demethylase lysine-specific demethylase 3A; KDM5A, histone H3K4 demethylase lysine-specific demethylase 5A; KDM5B, histone H3K4 demethylase lysine-specific demethylase 5B; PRMT6, protein arginine methyltransferase 6; SETD8, SET domain containing 8 or lysine methyltransferase 5A.

**Table 3 jox-15-00118-t003:** Impact of environmental xenobiotics on ncRNA expression.

Xenobiotic	Effects on ncRNAs	References
**Arsenic**	Increase expression of lncRNA-p21	[219]
**Cadmium**	Increased expression of lncRNA-ENST00000446135	[220]
**Mixture of pesticides (chlormequat chloride, pirimiphos-methyl, glyphosate, tebuconazole, chlorpyrifos-methyl, and deltamethrin)**	Downregulation of miRNA-146a	[221]
** *p* ** **,*p’*-DDT**	Upregulation of miRNA-19b, miRNA-27a, miRNA-126, miRNA-190a, miRNA-193b, and miRNA-378	[222]
** *o* ** **,*p’*-DDT**	Upregulation of miRNA-126, miRNA-190b, miRNA-193b, miRNA-324, miRNA-342, miRNA-378, and miRNA-423	[222]
	Downregulation of miRNA-190a	
**Benzo(*a*)pyrene**	Upregulation of miRNA-132	[223]
	Upregulation of miRNA-650	[224]
**Aflatoxin B1**	Upregulation of miRNA-19b, miRNA-19a, miRNA-34a, miRNA-99a, miRNA-190a, and miRNA-16	[225]
	Downregulation of miRNA-1307, miRNA-99b, and miRNA-100-5p	
**Aflatoxin B1**	Upregulation of lncRNA-H19	[226]
**Ochratoxin A**	Upregulation of miRNA-155	[182]
**Microcystin-LR**	Upregulation of miRNA-15b-3p	[227]
	Upregulation of miRNA-21, and miRNA-221	[149]
	Upregulation of miRNA-149-3p, miRNA-449c-5p, and miRNA-454-3p	[228]
	Downregulation of miRNA-122	[149]
	Downregulation of miRNA-500a-3p, miRNA-500a-5p, miRNA-500b-5p, and miRNA-4286	[228]
	Downregulation of miRNA-4521	[227]
**Okadaic acid**	Downregulation of miRNA-4492 and miRNA-4497	[229]

DDT, dichlorodiphenyltrichloroethane; lncRNA, long non-coding ribonucleic acid; miRNA, micro ribonucleic acid.

## Data Availability

Not applicable.

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
