# Peer review of "Environmental Xenobiotics and Epigenetic Modifications: Implications for Human Health and Disease"

_jox, 2025, doi:10.3390/jox15040118_

Round 1
Reviewer 1 Report
Comments and Suggestions for Authors
I think this is an interesting review type article, urgently filling the gaps of the literature as stated in the title: “Environmental xenobiotics and epigenetic modifications: Implications for human health and disease”. I have a few comments for the authors to further improve their work.
Figure 1. Hight resolution graph was needed. The current figure looks blurry with uneven color for the words. The abbreviation of bisphenol A: BPA can be added in the figure as stated in the text section.
Double check if the full name is showed before the abbreviation. Such as the full name of POPs was not shown.
The subtitles of the section of 4 is very long, the authors can shorten and concise them.
I suggest the authors to add the overall graphs for each section for section 4 through section 7 can help the reader to understand.
Additionally, a graphical abstract for the whole manuscript is highly recommended.
Author Response
REVIEWER 1
I think this is an interesting review type article, urgently filling the gaps of the literature as stated in the title: “Environmental xenobiotics and epigenetic modifications: Implications for human health and disease”. I have a few comments for the authors to further improve their work.
Authors’ response: We greatly appreciate the positive feedback of the Reviewer on the manuscript.
Specific Comments:
- Figure 1. Hight resolution graph was needed. The current figure looks blurry with uneven colour for the words. The abbreviation of bisphenol A: BPA can be added in the figure as stated in the text section.
Authors’ response: We greatly appreciate the Reviewer's comment and suggestion. In line with this, we have included figures in high-resolution TIFF format to avoid any problems related to figure quality.
We have also added the full names for BPA and 2,4-D in the figure to maintain consistency with other terms. We hope these revisions meet the Reviewer's expectations.
- Double check if the full name is showed before the abbreviation. Such as the full name of POPs was not shown.
Authors’ response: We greatly appreciate the Reviewer's suggestion. We have carefully reviewed the manuscript and ensured that all abbreviations, including "POPs" (persistent organic pollutants), are preceded by their full names at first mention.
- The subtitles of the section of 4 is very long, the authors can shorten and concise them.
Authors’ response: We greatly appreciate the Reviewer's suggestion. We have revised and shortened the subtitles in Section 4 to improve clarity and conciseness, while maintaining the intended meaning. We hope these revisions meet the Reviewer's expectations.
- I suggest the authors to add the overall graphs for each section for section 4 through section 7 can help the reader to understand.
Authors’ response: We greatly appreciate the Reviewer's suggestion. We have included three additional figures in the revised version of the manuscript to improve the Readers' understanding of the content. We hope the new figures meet the Reviewer’s expectations.
- Additionally, a graphical abstract for the whole manuscript is highly recommended.
Authors’ response: We greatly appreciate the Reviewer's suggestion. We have included a graphical abstract in the revised version of the manuscript to improve its clarity. We hope the graphical abstract meets the Reviewer's expectations.

Reviewer 2 Report
Comments and Suggestions for Authors
Critique:
Lines 31-36: These two sentences are redundant, i.e., they use a little different wording to express the same idea
Lines 34-37: These two sentences are redundant
Line 96: According to the given definition, oxygen and carbon dioxide are also environmental xenobiotics. Xenobiotics are rather "synthetic or natural chemical compounds that are foreign to a biological system, often introduced into the environment through human activity, and may affect ecosystems and human health."
Lines 117-118: This sentence implies that methylmercury is highly concentrated in seafood because it biomagnifies up the food chain, which makes no sense
Lines 120-125: Do you want to say that high mercury levels in those children were due to seafood consumption (see previous sentences in this paragraph), is it correct?
Line 127: What are groups 2B and 1? Is it good or bad to belong to these groups?
Lines 140-142: It says that daily intake of cadmium is estimated to be 30–50 μg. Is it about Netherlands? If so, to get such a dose one needs to drink 300-500L of water per day (10L in Sweden).
Figure 1: The major exposure routs are drinking, eating, breathing, or surface contact. “Industrial pollution” or “vehicle emissions” are xenobiotic sources rather than exposure routs
Line 162: Provided BPA concentration in food at 0.48 to 1.6 g/kg of body weight is a bit too high (by a few orders of magnitude)
Line 169: Was this concentration of BPA 450 ug calculated per cubic meter of human body?
Line 172: To the best of my knowledge, phthalates are liquids. How can a liquid be “elastic and flexible”?
Line 176: What are these “biological samples from the general population”? Please be more specific
Line 305: Authors need to call this “nitrogenous base” by its name, as in Fig.2
Line 315: DNMT2 does not methylate DNA. DNMT3L by itself has no DNA methyltransferase activity
Line 319: “(CpG) dinucleotides, which are found across CpG Islands in somatic cells” – are they not found in non-somatic cells?
Line 320: “In normal mammalian somatic cells, over 98%...” – what do you mean by “normal”? For example, in somatic embryonic cells, DNA methylation is erased at some point. Are these cells “normal” or “abnormal”?
Figure 2: In the bottom left cartoon representing methylated DNA, CH3 group is attached to G, explain.
Line 328: “potentially resulting in harmful health effects” – what do you mean by “potentially”? In the above chapters, you claimed that exposure to xenobiotics is harmful.
Line 362: “using acetyl coenzyme A as a co-factor” - in the reaction of acetylation of histones, acetyl coenzyme A serves as a substrate, not a co-factor
Table 2: Impact of environmental xenobiotics on histone modifications – Misleading title. This table also contains data on DNA methylation
Line 436: Misleading citation. In reference 120, it has not been shown that hypomethylation of those CpG sites “facilitates tumorigenesis”. This work established an association between the events, not a causal link
Line 456: Misleading citation. In reference 157, authors established an association of SNPs in CDH13 gene with the methylation of the CDH13 gene. By no means they have shown that “its downregulation through methylation contributes to tumorigenesis”
Line 494: Misleading citation. In reference 166, it has not been shown that the exposure-induced “disruption of chromatin regulation and DNA damage in the brain elevated the risk of AD”
Line 590: Please add citation(s) to the original Barker’s publication(s) on FOAD hypothesis
Line 662: Dutch famine was a famine, not an exposure to xenobiotics. If authors want to claim that there are similarities between the poor nutrition and xenobiotic exposures in utero, please provide more information (add a new chapter)
Line 667: How can these studies “suggest epigenetic transmission”?!
Line 670-671: How is it relevant to epigenetic transmission?
Line 711: It says that “These findings highlight the potential of HDAC inhibitors to act as epigenetic modulators “. No, these studies do not support this statement. Also, these studies do not show that xenobiotic-induced epigenetic changes are reversible. They simply show that xenobiotic-induced adverse outcomes can be partially reversed by an epigenetic inhibitor, highlighting the role of epigenetic processes.
Author Response
REVIEWER 2
Specific Comments:
- Lines 31-36: These two sentences are redundant, i.e., they use a little different wording to express the same idea.
Authors’ response: We greatly appreciate the Reviewer's comment. We have revised the two sentences to avoid redundancy.
In the revised manuscript, it reads as follows:
“Exposure to these compounds during critical periods, such as embryogenesis and early postnatal stages, can induce long-lasting epigenetic alterations that increase susceptibility to diseases later in life. Furthermore, modifications to the gamete epigenome can potentially lead to effects that persist across generations (transgenerational effects)”.
We hope this revision meets the Reviewer's expectations.
- Lines 34-37: These two sentences are redundant.
Authors’ response: We greatly appreciate the Reviewer's comment. We have revised the two sentences into a single sentence to avoid redundancy.
In the revised manuscript, it reads as follows:
“Furthermore, modifications to the gamete epigenome can potentially lead to effects that persist across generations (transgenerational effects)”.
We hope this revision meets the Reviewer's expectations.
- Line 96: According to the given definition, oxygen and carbon dioxide are also environmental xenobiotics. Xenobiotics are rather "synthetic or natural chemical compounds that are foreign to a biological system, often introduced into the environment through human activity, and may affect ecosystems and human health".
Authors’ response: We greatly appreciate the Reviewer's suggestion. We have revised the sentence in accordance with the Reviewer's suggestion.
In the revised manuscript, it reads as follows:
“Environmental xenobiotics are synthetic or natural chemical compounds that are foreign to a biological system, often introduced into the environment through human activity, and may affect ecosystems and human health”.
We hope this revision meets the Reviewer's expectations.
- Lines 117-118: This sentence implies that methylmercury is highly concentrated in seafood because it biomagnifies up the food chain, which makes no sense.
Authors’ response: We greatly appreciate the Reviewer's comment. We have revised the sentence to transmit the intended meaning more accurately.
In the revised manuscript, it reads as follows:
“One of its most toxic forms, methylmercury, accumulates in aquatic organisms and biomagnifies up the food chain, resulting in high concentrations in top predators such as seafood species”.
We hope this revision meets the Reviewer's expectations.
- Lines 120-125: Do you want to say that high mercury levels in those children were due to seafood consumption (see previous sentences in this paragraph), is it correct?
Authors’ response: We greatly appreciate the Reviewer's comment. In fact, we did not intend to correlate the increased levels of mercury found in children with seafood consumption. Therefore, to avoid misinterpretation, we have revised the paragraph to improve the transition between these distinct ideas.
In the revised manuscript, it reads as follows:
“Hg is a naturally occurring heavy metal, with a global production estimated at approximately 2,300 tonnes per year [38]. One of its most toxic forms, methylmercury, accumulates in aquatic organisms and biomagnifies up the food chain, resulting in high concentrations in top predators such as seafood species. Individuals who consume large amounts of seafood can accumulate methylmercury in their bodies and experience its harmful health effects [39].
More generally, Hg exposure remains a public health concern. According to the World Health Organization (WHO), an estimated 310,000 to 630,000 children in the United States of America (USA) were born with blood Hg levels exceeding 5.8 µg/L, placing them at risk of adverse developmental outcomes [40]. Similarly, in Europe, it is estimated that around 200,000 children exceed the WHO-recommended limit of 2.5 µg/g for Hg concentration in hair [41]”.
We hope this revision meets the Reviewer's expectations.
- Line 127: What are groups 2B and 1? Is it good or bad to belong to these groups?
Authors’ response: We greatly appreciate the Reviewer's questions. We have revised the sentence to clarify the meanings of Groups 1 and 2B.
In the revised manuscript, it reads as follows:
“Organic As has been placed in Group 2B (possibly carcinogenic to humans), while inorganic As is classified in Group 1 (carcinogenic to humans) by the International Agency for Research on Cancer (IARC) [42]”.
We hope this clarification meets the Reviewer's expectations.
- Lines 140-142: It says that daily intake of cadmium is estimated to be 30–50 μg. Is it about Netherlands? If so, to get such a dose one needs to drink 300-500L of water per day (10L in Sweden).
Authors’ response: We greatly appreciate the Reviewer's comment. We have revised the sentences to avoid misinterpretations and clarify the ideas.
In the revised manuscript, it reads as follows:
“In Europe, Cd concentrations in drinking water were found to be 5 µg/L in Sweden, while in the Netherlands, values ranged between 0.1 and 0.2 µg/L [48]. Although Cd exposure from drinking water is relatively low, the average total daily intake of Cd from all sources (including food, water, and airborne particles) is estimated to be 30–50 µg, which is associated with a range of health issues [45]”.
We hope this clarification meets the Reviewer's expectations.
- Figure 1: The major exposure routs are drinking, eating, breathing, or surface contact. “Industrial pollution” or “vehicle emissions” are xenobiotic sources rather than exposure routs.
Authors’ response: We greatly appreciate the Reviewer's suggestion. We have updated the subtitles in Figure 1 from “Exposure routes” to “Sources and exposure routes” to better reflect the content. We hope this alteration meets the Reviewer's expectations.
- Line 162: Provided BPA concentration in food at 0.48 to 1.6 g/kg of body weight is a bit too high (by a few orders of magnitude).
Authors’ response: We greatly appreciate the Reviewer's comment. In fact, where it reads “0.48 to 1.6 g/kg”, it should read “0.48 to 1.6 µg/kg”. We have corrected this mistake in the revised manuscript.
- Line 169: Was this concentration of BPA 450 ug calculated per cubic meter of human body?
Authors’ response: We greatly appreciate the Reviewer's question. According with the original study, workers from the factories were exposed to BPA at the mean personal airborne levels of 450 µg/m3. We have revised the sentence to avoid misinterpretations and clarify the idea.
In the revised manuscript, it reads as follows:
“In the case of occupational exposure, a study conducted by He and colleagues [57] observed that workers involved in BPA synthesis were more exposed to the compound, with an average personal airborne concentration of 450 µg/m³”.
We hope this clarification meets the Reviewer's expectations.
- Line 172: To the best of my knowledge, phthalates are liquids. How can a liquid be “elastic and flexible”?
Authors’ response: We greatly appreciate the Reviewer's comment. We have revised the sentence to avoid misinterpretations and clarify the idea.
In the revised manuscript, it reads as follows:
“Phthalates are hydrophobic compounds widely used in plastic manufacturing as plasticizers to enhance the elasticity and flexibility of plastics [58]”.
We hope this clarification meets the Reviewer's expectations.
- Line 176: What are these “biological samples from the general population”? Please be more specific.
Authors’ response: We greatly appreciate the Reviewer's suggestion. We have revised the sentence to make the idea clearer and more specific.
In the revised manuscript, it reads as follows:
“Several studies monitoring human exposure have concluded that di-(2-ethylhexyl) phthalate (DEHP) metabolites were found in 75–90% of urine samples from randomly selected individuals in the general population [60, 61]”.
We hope this clarification meets the Reviewer's expectations.
- Line 305: Authors need to call this “nitrogenous base” by its name, as in Fig.2.
Authors’ response: We greatly appreciate the Reviewer's suggestion. We have revised the sentence to explicitly name the nitrogenous base as cytosine.
In the revised manuscript, it now reads:
“DNA methylation is a well-established epigenetic mechanism that involves the addition of a methyl group to a cytosine [118] (Figure 2)”.
We hope this alteration meets the Reviewer’s expectations.
- Line 315: DNMT2 does not methylate DNA. DNMT3L by itself has no DNA methyltransferase activity.
Authors’ response: We greatly appreciate the Reviewer's comments. We have revised the paragraph to ensure the accuracy of the information.
In the revised manuscript, it reads as follows:
“DNA methylation is mainly carried out by a group of proteins called deoxyribonucleic acid methyltransferases (DNMTs), which include DNMT1 (responsible for maintaining methylation patterns during DNA replication), and DNMT3A and DNMT3B (involved in de novo methylation) [126, 127]. Although DNMT2 was initially classified as a DNA methyltransferase, it primarily methylates tRNA rather than DNA [128]. DNMT3L, on the other hand, lacks catalytic activity but serves as a regulatory co-factor that enhances the activity of DNMT3A and DNMT3B [126, 127].”.
We hope these clarifications meets the Reviewer's expectations.
- Line 319: “(CpG) dinucleotides, which are found across CpG Islands in somatic cells” – are they not found in non-somatic cells?
Authors’ response: We greatly appreciate the Reviewer's question. In fact, CpG islands are found in both somatic and non-somatic cells. Therefore, we have revised the sentence to avoid any misinterpretation.
In the revised manuscript, it reads as follows:
“5mC is particularly abundant in cytosine-phosphate-guanine (CpG) dinucleotides, which are concentrated in CpG Islands found throughout the genome [129]”.
We hope this clarification meets the Reviewer's expectations.
- Line 320: “In normal mammalian somatic cells, over 98%...” – what do you mean by “normal”? For example, in somatic embryonic cells, DNA methylation is erased at some point. Are these cells “normal” or “abnormal”?
Authors’ response: We greatly appreciate the Reviewer's questions. The term “normal” was intended to refer to differentiated somatic cells. However, we have revised the sentence to improve clarity and avoid any potential misinterpretation.
In the revised manuscript, it reads as follows:
“In mammalian somatic cells, over 98% of genome methylation occurs at CpG dinucleotides [130]. By contrast, in embryonic stem cells (ESCs), up to 25% of DNA methylation occurs at non-CpG sites [131, 132]. Moreover, both ESCs and early embryonic (somatic) cells undergo dynamic DNA methylation changes as part of normal epigenetic reprogramming [133]”.
We hope this clarification meets the Reviewer's expectations.
- Figure 2: In the bottom left cartoon representing methylated DNA, CH3 group is attached to G, explain.
Authors’ response: We greatly appreciate the Reviewer's comment. We have revised Figure 2 to correct the mistake.
- Line 328: “potentially resulting in harmful health effects” – what do you mean by “potentially”? In the above chapters, you claimed that exposure to xenobiotics is harmful.
Authors’ response: We greatly appreciate the Reviewer's point of view. We used the term “potentially” to acknowledge that while many studies have shown associations between xenobiotic exposure and altered DNA methylation patterns, the direct causal link to harmful health effects can vary depending on factors such as exposure levels, duration, and individual susceptibility. Thus, to avoid ambiguity, we have revised the sentence to better reflect this idea.
In the revised manuscript, it reads as follows:
“Some studies have demonstrated that exposure to environmental xenobiotics induces alterations in DNA methylation patterns that are associated with adverse health effects”.
We hope this clarification meets the Reviewer's expectations.
- Line 362: “using acetyl coenzyme A as a co-factor” - in the reaction of acetylation of histones, acetyl coenzyme A serves as a substrate, not a co-factor.
Authors’ response: We greatly appreciate the Reviewer's comment. Indeed, acetyl coenzyme A functions as a substrate, not a cofactor, in the histone acetylation reaction. We have revised the sentence accordingly to accurately reflect the role of acetyl coenzyme A.
In the revised manuscript, it reads as follows:
“HATs catalyse the transfer of an acetyl group to the ε-amino group of lysine side chains, with acetyl coenzyme A serving as the acetyl donor substrate [150]”.
We hope this clarification meets the Reviewer's expectations.
- Table 2: Impact of environmental xenobiotics on histone modifications – Misleading title. This table also contains data on DNA methylation.
Authors’ response: We greatly appreciate the Reviewer's comment. By mistake, we included data related to DNA methylation in Table 2, which was supposed to contain only data related to histone modifications. Therefore, we have moved the information related to DNA methylation to Table 1.
We hope this alteration meets the Reviewer's expectations.
- Line 436: Misleading citation. In reference 120, it has not been shown that hypomethylation of those CpG sites “facilitates tumorigenesis”. This work established an association between the events, not a causal link.
Authors’ response: We greatly appreciate the Reviewer's comment. We fully agree that a causal link was not established, only the observation of CpG site hypomethylation. Therefore, we have revised the sentence to transmit the correct idea. Reference 120 has been renumbered as reference 137 in the revised manuscript.
In the revised manuscript, it reads as follows:
“Long-term exposure to heavy metals has been shown to promote hypomethylation of CpG sites in key oncogenes like nuclear factor kappa B subunit 1 (NFKB1) [137].
We hope this clarification meets the Reviewer's expectations.
- Line 456: Misleading citation. In reference 157, authors established an association of SNPs in CDH13gene with the methylation of the CDH13gene. By no means they have shown that “its downregulation through methylation contributes to tumorigenesis”.
Authors’ response: We greatly appreciate the Reviewer's comment. We agree that reference 157 (174 in the revised manuscript) was not the most appropriate to support that idea. Therefore, we have revised the paragraph and added new references that better support the ideas discussed.
In the revised manuscript, it reads as follows:
“In human bronchial epithelial cells, environmental carcinogens like benzo(a)pyrene diol epoxide, a metabolite of tobacco smoke and urban air pollution, in-creased expression of DNMTs and caused hypermethylation-dependent downregulation of cadherin 13 (CDH13) [177]. CDH13 is considered an anti-oncogene, and its downregulation through methylation has been implicated in the initiation and pro-gression of different types of cancer [178, 179]”.
We hope these alterations meet the Reviewer's expectations.
- Line 494: Misleading citation. In reference 166, it has not been shown that the exposure-induced “disruption of chromatin regulation and DNA damage in the brain elevated the risk of AD”.
Authors’ response: We greatly appreciate the Reviewer's comment. We agree that reference 166 (185 in the revised manuscript) does not establish an association between exposure to particulate air pollution and increased risk of Alzheimer’s disease. Therefore, we have revised the paragraph to transmit this idea more accurately.
In the revised manuscript, it reads as follows:
“Recent evidence indicates that exposure to particulate air pollution, particularly met-al-rich combustion- and friction-derived nanoparticles, reduced the levels of di and trimethylated histone H3 at lysine 9 (H3K9me2/me3) in the prefrontal white matter of young individuals living in highly polluted environments [185]. Importantly, these individuals also showed elevated levels of hyperphosphorylated tau and amyloid-β (Aβ) plaques, hallmarks of AD [185]. Although a link between histone hypomethylation and tau hyperphosphorylation or Aβ plaques formation has not been established, this raises the possibility that particulate air pollution-induced histone modifications may increase the risk of AD”.
We hope this clarification meets the Reviewer's expectations.
- Line 590: Please add citation(s) to the original Barker’s publication(s) on FOAD hypothesis.
Authors’ response: We greatly appreciate the Reviewer's suggestion. We have added the appropriate references to the work of Barker and colleagues that laid the foundation for the “foetal basis of adult disease” hypothesis.
We hope this alteration meets the Reviewer's expectations.
- Line 662: Dutch famine was a famine, not an exposure to xenobiotics. If authors want to claim that there are similarities between the poor nutrition and xenobiotic exposures in utero, please provide more information (add a new chapter).
Authors’ response: We greatly appreciate the Reviewer's comment. Our intention in including the famine study was to illustrate that transgenerational effects, whether associated with xenobiotics or not, occur in humans. However, to avoid misinterpretations and confounding associations between poor nutrition and xenobiotic exposure, we have removed this study from the revised manuscript.
We hope this revision clarifies the Reviewer’s concern.
- Line 667: How can these studies “suggest epigenetic transmission”?!
Authors’ response: We greatly appreciate the Reviewer's question. We agree that these studies do not clearly demonstrate the involvement of epigenetic mechanisms in excess body fat. However, given that obesity is widely accepted to result from a complex interplay of genetic, epigenetic, and environmental influences, it is likely that tobacco smoke-induced epigenetic changes may contribute to this outcome, although further investigation is required. Moreover, the fact that transgenerational effects were observed beyond the second generation often suggest involvement of epigenetic mechanisms, as genetic mutations alone cannot explain inheritance across multiple generations. Thus, although the study did not directly measure epigenetic marks, it provides important evidence supporting the possibility of epigenetic inheritance Accordingly, we have revised the paragraph to better contextualize this hypothesis.
In the revised manuscript, it reads as follows:
“Additionally, recent studies suggested that the descendants (F1 to F3) of males exposed to tobacco smoke before puberty show increased body fat during childhood, adolescence, and early adulthood (Figure 6C) [257, 258]. As this is among the first evidence of xenobiotic-induced transgenerational effects beyond the second generation, it raises the possibility of epigenetic involvement, as the original idea that genetic mutations can fully explain heritability is no longer accepted [259, 260]. Moreover, given that obesity is widely accepted to result from a complex interplay of genetic, epigenetic, and environmental influences [261], it is likely that tobacco smoke-induced epigenetic changes may contribute to this outcome, although further investigation is required”.
We hope this clarification meets the Reviewer's expectations.
- Line 670-671: How is it relevant to epigenetic transmission?
Authors’ response: We greatly appreciate the Reviewer's question. The observation of transgenerational effects beyond the second generation often suggests the involvement of epigenetic mechanisms, as the original idea that genetic mutations can fully explain heritability is no longer acceptable. Therefore, we have revised the sentence to raise this hypothesis, although it is not yet conclusive.
In the revised manuscript, it reads as follows:
“As this is among the first evidence of xenobiotic-induced transgenerational effects be-yond the second generation, it raises the possibility of epigenetic involvement, as the original idea that genetic mutations can fully explain heritability is no longer accepted [259, 260]”.
We hope this clarification meets the Reviewer's expectations.
- Line 711: It says that “These findings highlight the potential of HDAC inhibitors to act as epigenetic modulators “. No, these studies do not support this statement. Also, these studies do not show that xenobiotic-induced epigenetic changes are reversible. They simply show that xenobiotic-induced adverse outcomes can be partially reversed by an epigenetic inhibitor, highlighting the role of epigenetic processes.
Authors’ response: We greatly appreciate the Reviewer's comments. We agree with the Reviewer’s point of view and have revised the sentence to make the idea clearer.
In the revised manuscript, it reads as follows:
“These findings suggest that HDAC inhibition can partially reverse ethanol-induced neurobehavioral alterations, highlighting the involvement of epigenetic mechanisms in these outcomes, rather than proving the reversibility of specific epigenetic changes”.
We hope this clarification meets the Reviewer's expectations.

Reviewer 3 Report
Comments and Suggestions for Authors
The manuscript provides a clear, detailed, and well-structured overview of the relationship between environmental xenobiotics and epigenetic modifications, emphasizing how compounds such as heavy metals, pesticides, endocrine-disrupting chemicals, and air pollutants can induce epigenetic alterations with potential transgenerational effects on human health. The work is well-executed, with informative figures and tables that effectively support and enhance the narrative. One aspect that could have been explored in greater depth is the role of nano- and microplastics as carriers of xenobiotics. Additionally, nanoplastics themselves may exert direct epigenetic effects, as previously demonstrated (doi: 10.3390/ijms241411379). Overall, I believe the manuscript is of high quality and would only require minor revision before considering it for publication in JoX.
Author Response
REVIEWER 3
The manuscript provides a clear, detailed, and well-structured overview of the relationship between environmental xenobiotics and epigenetic modifications, emphasizing how compounds such as heavy metals, pesticides, endocrine-disrupting chemicals, and air pollutants can induce epigenetic alterations with potential transgenerational effects on human health. The work is well-executed, with informative figures and tables that effectively support and enhance the narrative. Overall, I believe the manuscript is of high quality and would only require minor revision before considering it for publication in JoX.
Authors’ response: We greatly appreciate the positive feedback of the Reviewer on the manuscript.
Specific Comments:
- One aspect that could have been explored in greater depth is the role of nano- and microplastics as carriers of xenobiotics. Additionally, nanoplastics themselves may exert direct epigenetic effects, as previously demonstrated (doi: 10.3390/ijms241411379).
Authors’ response: We greatly appreciate the Reviewer's comments. In line with this pertinent point, we have added section 2.5 to the manuscript, focusing on nano- and microplastics as environmental xenobiotics. In this section, we also discuss how nano- and microplastics can adsorb and transport other environmental pollutants, enhancing their bioavailability and toxicity. Furthermore, as suggested, we have incorporated recent findings to highlight emerging evidence of their ability to directly alter DNA methylation.
In the revised manuscript, it reads as follows:
“2.5. Nano- and microplastics
Nano- and microplastics (NPs and MPs, respectively) are plastic particles smaller than 1 μm and 5 mm, respectively [105, 106]. They originate either from the fragmentation of larger plastic debris (secondary particles) or are manufactured intentionally for use in industrial applications, cosmetics, and textiles (primary particles) (Figure 1E) [107]. Due to their widespread use and environmental persistence, NPs and MPs have become ubiquitous pollutants, being detected in drinking water [108], food products [109], air [110], and even human tissues, including lungs [111], and blood [112]. As such they are classified as environmental xenobiotics.
On the other hand, NPs and MPs can adsorb and transport other environmental contaminants, such as heavy metals [113], pesticides [114], and POPs [115], enhancing their mobility and bioavailability. Studies have demonstrated that NPs can disrupt cellular homeostasis, leading to cytotoxic and genotoxic effects [116]. While their health impacts remain under investigation, growing evidence indicates that long-term expo-sure to NPs/MPs can induce epigenetic alterations [117]”.
We hope this alteration meets the Reviewer's expectations.

Round 2
Reviewer 1 Report
Comments and Suggestions for Authors
The authors addressed all my questions.